# A numerical model of microplastic erosion, transport, and deposition for fluvial systems

John J. Armitage[1] and Sébastien Rohais[1]

[1]IFP Energies Nouvelles

**Correspondence:** John J. Armitage (john-joseph.armitage@ifpen.fr)

**Abstract.** Rivers are the primary pathway of microplastic pollution from source to the eventual sink in the marine environment. However, like sediments, microplastics will become trapped within the fluvial system as they make their way from source-to-sink. Therefore there is the potential that rivers are an important reservoir of microplastic pollution globally. To explore the transport of microplastic through the fluvial system we develop a reduced complexity model of microplastic erosion, transport, and deposition that builds on methods developed for the transport of sediment. We apply this model to the river Têt, France, where there has been punctual monitoring of the flux of microplastic at the outlet. We find that the reduced complexity model captures the observed quantity of microplastic under reasonable assumptions of the relationship between microplastic sources and population density. The model that best matches observed fluxes of microplastic at the outlet of the Têt river requires between 1 and 10 ppm volume concentration of microplastic per $200\times200$ m in the top half a meter of soil. The microplastic of grain size 300 $\mu$m then travels within the river network with a settling velocity of the order of $10^{-4}$ m/sec. The model results imply that a large proportion of microplastic will become entrained within the sediments along the fluvial system. This model is a first step in assessing where to sample for microplastic pollution within fluvial systems and points to regions susceptible to microplastic pollution.

## 1 Introduction

It is not controversial to state that there is an increasing awareness of the threat of plastic pollution on the natural environment and public health. Plastics are a highly durable material and cheap to produce. They are used in nearly all industrial processes and have entered the atmosphere, hydrosphere, lithosphere, and biosphere. Plastics once used are either recycled, incinerated, or dumped. It was estimated that roughly 9% of plastics created are recycled, and about 12% are incinerated (Geyer et al., 2017). What is left is discarded either in landfill, in open pits as mismanaged waste, or enters the environment through waste water treatment. Estimates of total plastic show an exponential growth in production. In 2015 it was estimated that 407 Mt of primary plastic (virgin plastic created from raw materials) entered into a use phase, while 302 Mt left the use-phase (Geyer et al., 2017). Of the stock of plastic that is estimated to have left use, roughly 80% will enter landfill or be dumped in open pits. Plastic that is not contained within managed landfill is termed mismanaged plastic waste (MPW) and it was estimated that in 2015 there was between 65 and 99 Mt of MPW (Lebreton and Andrady, 2019). Of this, estimates are that somewhere between 0.8 and 4 Mt per year enters the ocean via the river network (Schmidt et al., 2017; Lebreton and Andrady, 2019; Meijer et al., 2021). The

estimates of plastic waste that enters the ocean differ on the assumptions for the spatial distribution of MPW and the impact of climate on river run-off on the delivery of plastic to the oceans. This means that the importance of small river catchments versus large rivers is still uncertain. This uncertainty is demonstrated by, for example, a local study of microplastic contamination within the Mersey and Irwel rivers in England. These two small rivers were found to contain roughly half a million particles of microplastic per meter, which makes them one of the most polluted rivers globally (Hurley et al., 2018). This watershed however does not suffer from open tipping of plastic, and there is no strong evidence for mismanaged water treatment. Therefore, it is possible that either significantly more plastic than previously thought is stored within rivers (van Emmerik et al., 2022), or significantly more plastic than estimated enters the oceans.

The challenge is that currently there is not sufficient data to distinguish between the quantity of river and ocean microplastic storage, and furthermore current models do not take into account the transport and deposition of microplastic particles down system. The box model developed by Sonke et al. (2022) for a global mass balance of microplastic treats river transport only as an input to the ocean sinks and reaches the conclusion that a considerable quantity of microplastics accumulate in the deep ocean. On a global scale it has been observed that the quantity of microplastic that enters the rivers is related to the population density (Weiss et al., 2021). However, global studies treat river catchments as single point discharge points and do not include permanent and temporary storage of microplastic within the catchment. In this study we wish to model the transport of microplastic within the rivers to capture the potential for microplastic to become deposited within the fluvial system en route to the ocean basin. The goal is to develop a tool that may help understand where temporary or permanent storage of microplastics will occur within river systems.

## 2 Brief review of existing models for landscape-scale transport of microplastics

There is a lack of numerical models for the transport of microplastics in fluvial systems, as can be seen in the compilation of models within Waldschläger et al. (2022). Within this small collection, the process-based model `INCA-Contaminants` was developed with the aim to model the transport of microplastics (Nizzetto et al., 2016). This model divides the river catchment into reaches and within each reach assumes sediment is transported along the reach with the water flow and with a fall velocity. `INCA-contaminants` was used to explore the potential microplastic transport within the River Thames, UK. Unfortunately, there are no observations for microplastics within the water column or within the sediments for any comparison between the model prediction and the natural river system. In effect, the model treats microplastic as a particularly light sediment, yet this assumption has not been tested. The model `INCA-contaminats` is a slight modification on `INCA-seds` (Lazar et al., 2010). These models solve for the transport of particles by reducing the spatial dimensions to one dimension and splitting the catchment into reaches and modelling the various input pathways to each reach (Lazar et al., 2010). The reach approach has also been used in a second model that captures the advection and dispersion of microplastic down system (De Arbeloa and Marzadri, 2023), rather than the more simple settling approach of `INCA-seds`. Both models reduce the two dimensional catchment topology to a series of one dimensional ordinary differential equations and as such the lateral spatial variability in sediment or microplastic deposition cannot be captured. Instead the focus is on capturing the flux of microplastic as it enters at point locations and leaves

the catchment at a particular gauging station or other point of interest (De Arbeloa and Marzadri, 2023; Nizzetto et al., 2016).
This means the long-term storage of microplastic within the floodplains or other three-dimensional natural barriers cannot be captured.

A different approach would be to model the full problem of the flow of surface water, as is taken to model particles of car tires (Unice et al., 2019). These particles are assumed to travel with runoff and hence a hydrological model, `DELFT3D-WAQ`, is used to model the freshwater flow. Based on the flow pathways and estimates of removal of the particles from these flow pathways due to water treatment and deposition within the sediment, the amount of tire particles that remain in the environment can be estimated. This sort of model is essentially a post-processing of a hydrological model that solves for the shallow water equations within the terrestrial environment. As such it might not be appropriate for exploring how microplastics get transported, stored, and remobilized within the fluvial environment.

In estuaries, the transport of microplastic has been modelled at a similar level of complexity as the tire particles discussed above. With a focus on the Chesapeake Bay, the transport of microplastic treated as a trace particle in the regional ocean circulation model applied to the estuary was used to predict where microplastics would be distributed in the water column (López et al., 2021). This approach however excludes the potential for deposition of the microplastic and is limited to regions where there are large bodies of water, such as lakes and estuaries. As such it is difficult to adapt this to microplastic transport within rivers, where the overland flow of water cannot be captured with the same assumptions behind oceanographic circulation models.

As a compromise between the reduced dimension model and the full Navier-Stokes problem, the transport of microplastics could be captured using reduced complexity models as applied to tracking contaminated sediment or to study the source-to-sink pathway of sediment released from earthquakes (Coulthard and Macklin, 2003; Xie et al., 2022). These models use the empirical transport equations for sediments, such as that developed by Wilcock and Crowe (2003) to link the water flux to sediment flux to capture the transport of multiple grain sizes down system (Coulthard et al., 2013). Subsequently, certain grains are tracked as they flow through the system. In these examples, the landscape evolution model is used to track sediments and not microplastics, however there is the potential in this approach for tracking microplastic as it migrates down the fluvial system. The advantage is that the spatial distribution of deposition can be captured, in particular deposition in floodplains, levees and the riverbanks. For example, from modelling lead contamination in sediments it was found that avulsions within the depositional environment lead to a significant reworking of previous deposits and a release of contaminants into the watershed (Coulthard and Macklin, 2003). Given that microplastics are found within riverbank deposits, avulsions would have a similar impact on microplastic contamination.

## 3  Modelling microplastic as a sediment

We suggest to model microplastics in the same way that sediment transport is modelled within landscape evolution models and stratigraphic models. That is we make the same simplifying assumptions: (1) Microplastic is transported with the water and falls out of suspension with a characteristic velocity, $v_{drop}$ (Figure 1). (2) Microplastic is eroded from the river bed when the shear

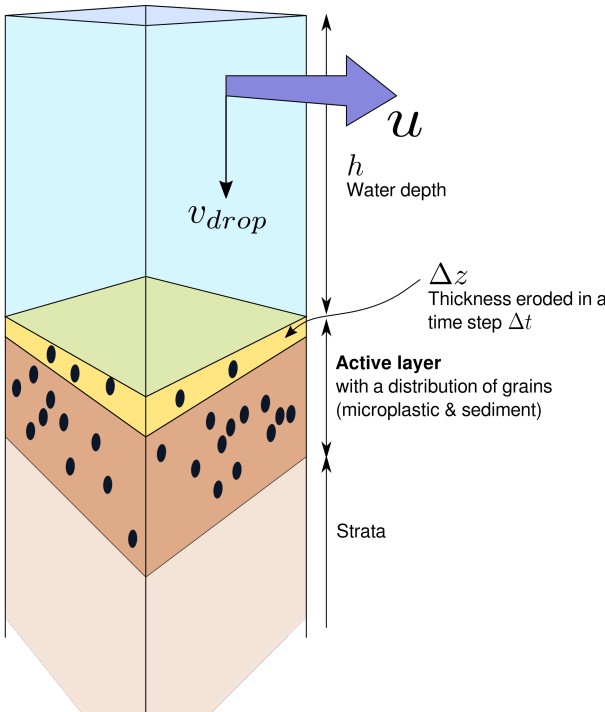

**Figure 1.** Diagram of the basic transport assumptions for the inclusion of microplastic within a model of sediment transport. Microplastic is transported within the water column of height $h$ and velocity $u$. Microplastic otherwise resides within the active layer of sediment as particles that can be eroded and put into suspension. Once in suspension they fall with a characteristic fall velocity $v_{drop}$.

stress exceeds a critical threshold that is dependent on the mix of grain sizes in the bed (Figure 1). (3) That there exists an active layer that participates within erosion and deposition (Figure 1). The active layer contains a distribution of sediment grains sizes along with the microplastic, and the median sediment grain size of this distribution will alter the quantity of microplastic eroded due to a microplastic hiding function. Microplastics come in a complex array of shapes, sizes and densities, as do sediments, however these simplifying assumptions allow a reduction in the complexity and have proven to be sufficient to capture many aspects of sediment transport within landscape evolution models and stratigraphic models.

### 3.1 Water flux

To include microplastics within a sediment transport model we build upon the framework of `CAESAR-Lisflood` (Coulthard et al., 2013), where the flow of water is solved for assuming the diffusive-wave approximation to the shallow water equations, and that the microplastic is eroded from the bed above a critical threshold. The full Saint-Venant shallow water equations can be written as,

$$\frac{\partial h}{\partial t} + \frac{\partial}{\partial x}(hu) = p \tag{1}$$

for the conservation of mass and,

$$\frac{\partial}{\partial t}(hu) + \frac{\partial}{\partial x}\left(hu^2 + g\frac{h^2}{2}\right) = -gh\frac{\partial z}{\partial x} - \tau \tag{2}$$

for the conservation of momentum. Where $h$ is the water depth, $u$ is the velocity, $p$ is the precipitation rate, $g$ is gravity, and $\tau$ is the friction term. We will use the same friction model as `Lisflood-FP` (Bates et al., 2010),

$$\tau = \frac{gn^2(hu)^2}{h^{7/3}} \tag{3}$$

where $n$ is Manning's roughness. In the diffusive wave assumption the $\partial/\partial x\left(hu^2\right)$ is assumed to be insignificant. Under this assumption the momentum balance reduces to,

$$\frac{\partial}{\partial t}(hu) + \frac{\partial}{\partial x}\left(g\frac{h^2}{2}\right) = -gh\frac{\partial z}{\partial x} - \tau \tag{4}$$

This reduced form of the shallow water equations can be solved for using semi-implicit or fully implicit schemes. The diffusive-wave approach ignores the advective terms and as such cannot capture the migration of a wave of water down the fluvial system. Furthermore, by not including the advective terms an instability in the solution can form as the flow accelerates down slope. One simplification to avoid such instabilities is to include a limit on the flux to keep the flux below a certain Froude number (Coulthard et al., 2013),

$$F = \frac{u}{\sqrt{gh}} \tag{5}$$

such that the flux limiter is given by,

$$q_{lim} = F_{lim}h\sqrt{gh} \tag{6}$$

where $F_{lim}$ is the maximum permissible Froude number for the flow and $q_{lim}$ is therefore the maximum water flux.

## 3.2 Microplastic transport

We will include microplastic as a grain fraction along with a selection of sediment grain fractions, where the input grain size distribution, $F_i$, is distributed over an active layer thickness, $z_a$, to give a thickness of each grain size class, $g_i$ (Van De Wiel et al., 2007). Here $i \in \{0, N\}$ and $N$ is the number of grain size classes (including both microplastic and sediment grains). For each grid point in the model, we store the fraction of grains for each class, from microplastic through silts to gravels. The distribution is then updated at each time step as the grains get transported down system by overland flow of water.

To estimate the flow of sediment from the flow of water there are a few empirical flow calculations that can be used, see Parker (2008) for a review. In the landscape evolution model `CAESAR-Lisflood`, the multi-grain model of Wilcock and Crowe (2003) is implemented. This model has the advantage that it can approximate the transport properties of multiple grain size fractions. It is however based on an extrapolation from laboratory experiments, and it is not straightforward to include microplastics within this model. We therefore chose to include microplastics separately using a Meyer-Peter and Müller law as parameters such as the hiding function have already been estimated for microplastics (Waldschläger and Schüttrumpf, 2019a).

For both sediment and microplastics we calculate the shear stress on the bed, $\tau_b$, via the Manning-Strickler empirical law, such that,

$$\tau_b = \rho_w C_f u^2 \tag{7}$$

where,

$$C_f = g n^2 h^{-1/3} \tag{8}$$

To calculate the microplastic flux, where we assume a single grain size $D_p$, we use the simple empirically derived model that above a shear stress, given by the dimensionless Shields number, the microplastic will be in motion. From the shear stress the dimensionless Shields number is given by,

$$\theta = \frac{\tau_b}{\rho s g D_p} \tag{9}$$

The threshold Shields number for microplastic can be derived from laboratory experiments where the hiding effect that sediment grains have on the microplastic is estimated as (Waldschläger and Schüttrumpf, 2019a),

$$\theta_{tp} = 0.5588 \theta_t \left( \frac{D_p}{D_m} \right)^{-0.503} \tag{10}$$

where $\theta_t$ is the threshold Shields number for the median grain size ($D_m$) of the distribution of grain sizes in the active layer. Subsequently, if the Shields number exceeds the threshold, the dimensionless flux per unit width is,

$$q^\star = F_p C_p (\theta - \theta_{tp})^{1.5} \tag{11}$$

where $F_p$ is the fraction of microplastic in the active layer and $C_p$ is a coefficient normally derived from rating curves. For sediments $C_p$ is estimated to be equal to 3.97 (Wong and Parker, 2006; Huang, 2010). A similarly large data set of catchment-scale observations does not exist for microplastic, but from laboratory experiments $C_p$ can be estimated to be of the order of 2.4 (Figure A1). The precise value of $C_p$ can be tuned for landscape scale model runs. From the Shields number the erosion flux of per unit width (in the 1D model units of m/sec) is,

$$E_p = q^\star \left( s g D_p^3 \right)^{1/2} \tag{12}$$

The loss of microplastic due to transport is then used to update the thickness of the active layer. Note that the active layer is equal to the sum of the thicknesses of all grain size fractions that make up the active layer from microplastic to silt, sand, and gravel.

Once the microplastic has left the bed it is transported within the water column. It is assumed that there is a maximum transport capacity and if that limit is already reached no more microplastic can enter the water column, even if the threshold Shields number is exceeded. The microplastic will subsequently fall out of suspension depending on its density. For simplicity we assume that the microplastic has a constant fall velocity, and the deposition of the microplastic can be given from the rate at which the microplastic settles out of the water column. The mass balance for microplastic is therefore given by,

$$\frac{\partial h_{susp}}{\partial t} + \frac{\partial}{\partial x} (h_{susp} u) = E_p - v_{drop} \left( \frac{h_{susp}}{h} \right) \tag{13}$$

Where $h_{susp}$ is the thickness microplastic particles in suspension (which cannot exceed $C_{susp} h$ where $C_{susp}$ is the dimensionless transport capacity of the water column), $E_p$ is the erosion rate from equation (12), and $v_{drop}$ is the fall velocity (Figure 1).

### 3.3 Sediment transport

As mentioned above, the transport of sediment from the active layer is calculated from the empirical law developed by Wilcock and Crowe (2003). The inclusion of this law follows the same methods as in `CAESAR-Lisflood`, and for completeness it is outlined here. From the shear stress the shear velocity is calculated as

$$u_\star = nug^{1/2}h^{-1/6} \tag{14}$$

The rate of erosion is then given by,

$$E_{s,i} = \frac{F_i u_\star^3 W_i^\star}{sg} \tag{15}$$

where $s$ is the specific gravity of the sediment and $W_i^\star$ is a power law function that is dependent on the grain size and shear stress. A reference shear stress for each grain size fraction is defined empirically as,

$$\tau_{r,i} = \rho g D_m \left(0.021 + 0.015 e^{-20F_s}\right) \left(\frac{D_i}{D_m}\right)^{0.67/\left(1+e^{(1.5-(D_i/D_m))}\right)} \tag{16}$$

where $F_s$ is the fraction of sand within the grain size distribution, $D_i$ is the grain size, and $D_m$ is the median grain size in the active layer. Depending on the magnitude of the ratio of shear stress to reference shear stress, the function $W_i^\star$ is given by,

$$\frac{\tau_b}{\tau_{r,i}} < 1.35, \quad W_i^\star = 0.002 \left(\frac{\tau_b}{\tau_{r,i}}\right)^{7.5} \tag{17}$$

or,

$$\frac{\tau_b}{\tau_{r,i}} \geq 1.35, \quad W_i^\star = 14 \left(1 - 0.0894 \left(\frac{\tau_b}{\tau_{r,i}}\right)^{-0.5}\right)^{4.5} \tag{18}$$

The finest grain size is treated as a suspended particle, and its transport and deposition follows from equation (13). For the bed load the particles are routed down slope (Coulthard et al., 2013).

### 3.4 Active layer and strata below

Following Van De Wiel et al. (2007) sediment layers are included, forming an active layer on the surface and then 9 *strata* layers below this (Figure 1). Each layer is defined to have an initial distribution of sediment grain sizes and a thickness (typically less than one meter). Erosion and deposition act on the active layer, which is the layer at the surface (Figure 1). The inclusion of the notion of an active layer allows us to include microplastics within this layer only. This will allow the modelling of how the microplastics in a contaminated layer get remobilized down the fluvial system. The active layer can be potentially significantly eroded or become excessively thick due to deposition. Therefore a rule is included to allow for material below to be added for the case of erosion, or for a new strata to be created in the case of deposition.

- Erosion: When the active layer is too thin the strata layer below is added to create a new active layer. The rule used is if the active layer thickness drops below 0.25 times its starting thickness then the strata layer below is added. This causes a

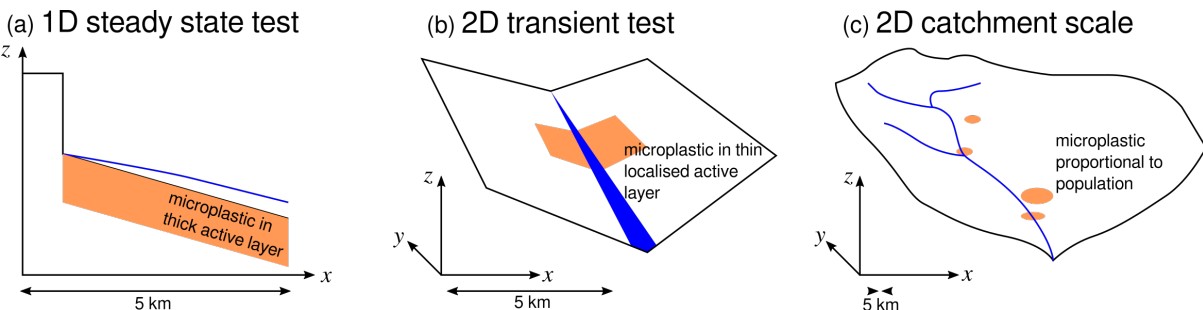

**Figure 2.** Diagrams of the three different test cases for the microplastic transport model. (a) A 1D steady state test where the active layer that contains microplastic is thick, 1 m thick, to avoid the exhaustion of the microplastic source. (b) A simple 2D transient case, where a thin, 0.1 m thick, region of microplastic is contained in the middle of a small hypothetical catchment. (c) An application of the model to the catchment scale, the lower catchment of the River Têt.

change in the proportion of grains, as upon adding the strata layer, the distribution of grains includes that from the layer below. This makes a mechanism that is similar to eroding away the cover of an alluvial bed and accessing the sediments below. Upon incorporation of the top strata layer a new bottom strata layer is added in the model.

- Deposition: When the active layer is too thick a new strata layer is created. In this case if the thickness of the active layer exceeds 1.5 times its initial thickness the active layer gets split into a new top strata layer and a new active layer that is 0.5 times the original active layer thickness. The new active layer has the same grain distribution as before and the new top strata layer has the distribution of grains with which it was created frozen in. Upon creation of the new top strata layer, the bottom strata is deleted in the model.

The thickness of the active layer can be somewhat arbitrary (Parker, 2008). Within the river bed it can be taken to be thin, and approximated from the largest grain size in transport. In a catchment scale landscape evolution model, where the model resolution is of grid sizes of 200 × 200 m, then the thickness of the active layer needs to encompass natural, agricultural, rural and urban land-use, and ridge tops, hillslopes and valley floors. In this study we vary the active layer thickness depending on the application of the model. In the subsequent sections we will first explore a 1D model to look at the steady state variability in microplastic transport, then we test a simple 2D setup, and finally a catchment scale test (Figure 2). For 1D model scenario tests, we define a thick active layer of 1 m so that microplastic supply is not exhausted with the aim of understanding the steady state behaviour (Figure 2a). For 2D model scenarios we keep the active layer thin, 0.1 m, to understand the transient behaviour of the model (Figure 2b). For the application to the Têt River system the model resolution is wider than an individual river channel, at 200 × 200 m. We therefore chose to set the active layer at 0.5 m (Figure 2c).

## 4 Transport of microplastic and sediment in 1D

To test the numerical stability and applicability of the simple microplastic transport model coupled to the sediment transport model we first reduce the complexity of the problem and explore the limits and sensitivity of the model assumptions in one dimension (Figure 2a). We solve for the flow of water using a implicit finite difference scheme and then update for the flow of sediment using a simple up-wind first order scheme. The shallow water equations are only valid for regions where there is water. The erosion of microplastic and sediment is only applied if the flow height of water is above 1 cm, as below 1 cm the flow is

minimal. This condition is to avoid the application of the transport laws to unrealistically small flows of water.

We initiate the 5000 m long model with a simple ramp-like topography that decreases from left to right, with a step increase in elevation at the right left side from 0 to 200 m distance to avoid the flow of water out of the left hand boundary (Figure 2a). The effective model length is therefore 4800 m. The active layer thickness is set to 1 m thick, and nine strata layers below are each 0.5 m thick. Precipitation rate is set to 0.01 m/hr for all cells along the domain except for those that are within the region of

220 high elevation on the right hand boundary, where it is set to zero. The high precipitation rate is used as water can only enter the 1D model from above, there is no runoff from the sides. The boundary conditions for the water flow are of zero flow on the left hand side, here there should be no water. On the right hand side we keep the hydraulic gradient constant across the boundary, assuming that the slope is likewise constant across the boundary. Testing of the model found that for slopes greater than 2° the water flux exceeds the Froude number limit (Equation (6)). We set the slope to 0.1° to keep the flow velocities low.

The model is initiated with a distribution of grain sizes, where there are seven sediment grain classes and one microplastic class. We test five different median grain sizes, $D_m$ of 0.11, 0.28, 0.56, 1.13, and 3.87 mm, where the median grain size is estimated assuming a continuous log-normal distribution between the two most abundant grain sizes within the distribution of sediment grains (Table A1). We also test five different microplastic grain sizes, $D_p$, from 1 $\mu$m to 1 mm and explore the impact of five microplastic fall velocities from $10^{-6}$ to $10^{-2}$. The large range of fall velocities is explored to cover the range of velocities

observed from laboratory experiments on microplastics and the full range of fall velocities for denser quartzite sedimentary grains (Waldschläger and Schüttrumpf, 2019b; Dietrich, 1982).

### 4.1 Steady state fluxes and model resolution

At steady state the water depth reaches a maximum of 0.06 m with a precipitation rate of 0.01 m/hr. With this small amount of water only a very small amount of microplastic is entrained into the water. The flow of water down this simple slope is not

trivial and the spatial resolution will impact the solution given. To explore numerically where the model is robust we first run a simple test with increasing spatial resolution, with 50, 200 and 500 cells to represent the 5000 m long domain, or equivalent to 100, 25, and 10 m cell sizes. A steady state water flux of 48 m$^2$/hr is reached for the models with a resolution of 25 and 10 m cells. Sediment and microplastic transport will slowly reduce as the model domain is eroded to a flat surface, however we find that as the model resolution increases the flux of microplastic out of the 1D slope converges towards the same trend

(Figure 3). Therefore, we will take a resolution of 25 m cells in the following analysis of the 1D model to explore how the grain size distribution impacts microplastic transport for this simple case.

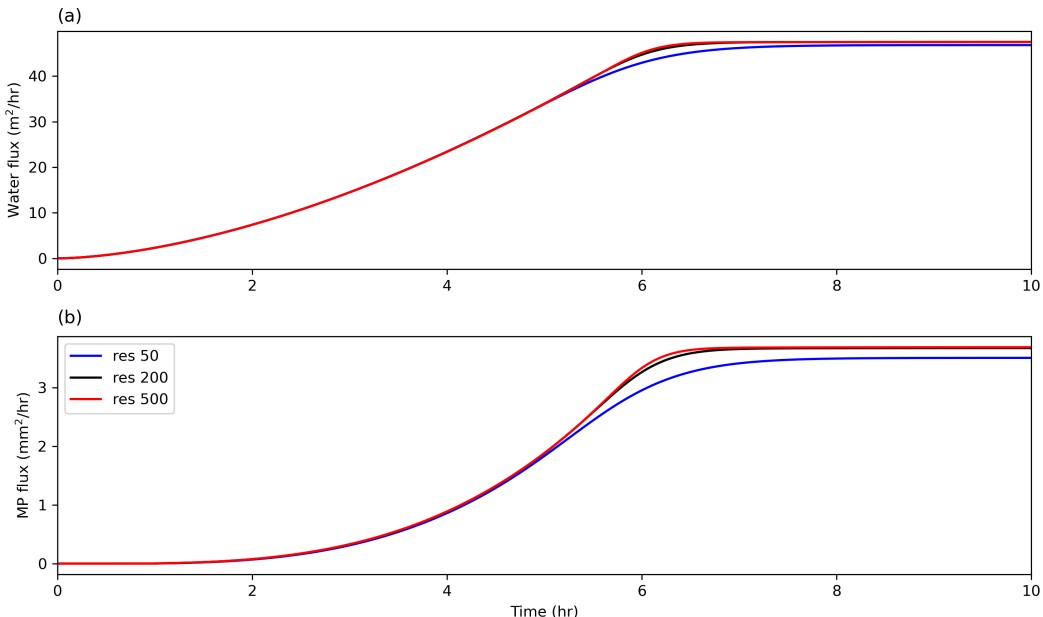

**Figure 3.** Water and microplastic flux out of the 1D model for different model resolutions. (a) Water flux and (b) microplastic (MP) flux for a resolution of 50, 200 or 500 cells, corresponding to a length of 100, 25 and 10 m cell length.

## 4.2    Impact of grain distribution and microplastic size

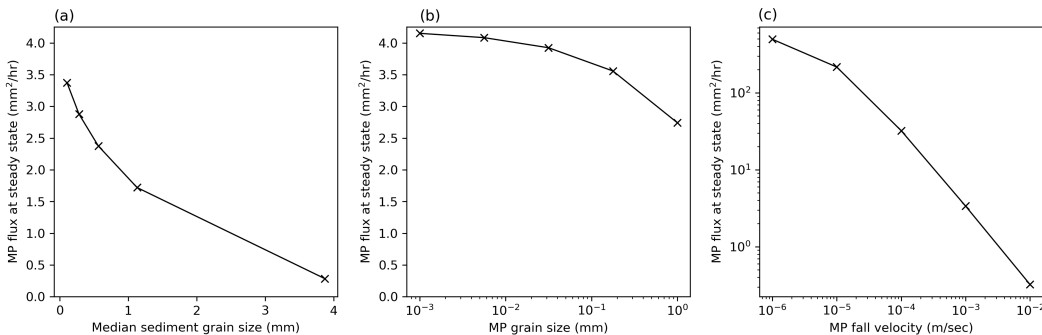

**Figure 4.** Steady state microplastic flux for different model conditions. (a) Steady state microplastic flux as a function of the median sediment grain size (microplastic grain size is 300 $\mu$m and fall velocity of $10^{-3}$ m/sec). (b) Steady state microplastic flux as a function of the microplastic grain size (median grain size is 110 $\mu$m and fall velocity of $10^{-3}$ m/sec). (c) Steady state microplastic flux as a function of the fall velocity of the microplastic. For all models the precipitation rate is 0.01 m/hr and the slope is 0.1° (microplastic grain size is 300 $\mu$m and the median grain size is 110 $\mu$m). For all models the precipitation rate is 0.01 m/hr and the slope is 0.1° .

We test the hiding impact of five different grain size distributions for the 7 grain size classes via the median grain size, $D_m$. The microplastic grain size is fixed at 300 $\mu$m and fall velocity is $10^{-3}$ m/sec. Increasing the median sediment grain size has

the impact of reducing the quantity of microplastic transported and this reduction is not linear, the effect is increased for small median grain sizes (Figure 4a). A small increase from a median grain size that is in the range of silt to fine sand (110 $\mu$m) to fine sand (280 $\mu$m) will reduce the microplastic flux by 12% (first two points in Figure 4a).

We also explore the impact of different microplastic grain sizes in the erosion model keeping the median grain size fixed at 110 $\mu$m and the fall velocity at $10^{-3}$ m/sec). Increasing the microplastic size reduces the quantity exported from the 1D slope

(Figure 4b). For example, relative to the model with a microplastic size of 300 $\mu$m a reduction to 100 $\mu$m increases the volume transported by 10% while increasing the grain size to 1 mm reduces the volume transported by 21%. This is to be expected from the definition of microplastic transport, where if we increase the grain size, the Shields number reduces and the threshold Shields number increases. The result is that for larger grains a smaller quantity is entrained.

Finally the fall velocity impacts the microplastic flux, as it changes the quantity in suspension. We vary the fall velocity

keeping the microplastic grain size fixed at 300 $\mu$m and the median grain size fixed at 110 $\mu$m. An order of magnitude reduction in the fall velocity causes an order of magnitude increase in microplastic flux (Figure 4c; note the logarithmic scale). Within the range of observed fall velocities for microplastics ($> 10^{-4}$ m/sec) the steady state flux of microplastic is as high as 31 mm$^2$/hr (Figure 4c). If the buoyancy of the microplastic, and processes such as saltation within transport reduce the rate of settling further then the steady state flux will increase to the order of 100's of mm/hr. Out of the three variables, the settling velocity

might have the largest impact.

## 5    Transport of microplastic in 2D - incorporation into CAESAR-Lisflood

The 1D model tests suggest that the microplastic transport model developed can be incorporated into the 2D landscape evolution model `CAESAR-Lisflood`, in this case the C++ version `HAIL-CAESAR` (Valters, 2023). The modified version of `HAIL-CAESAR` containing microplastic is currently a fork of the main `HAIL-CAESAR` repository (Armitage, 2023).

Microplastic is included as a grain size that is transported within the water column as described in equation (13). Building on the 1D model we first test the 2D model by including a source region of microplastic within a wedge-like topography (Figure 2b). The initial condition is of a topography that is like a tilted and folded piece of paper, where the slope along the axis of the valley is 1°. The initial condition includes a slight surface roughness of a maximum amplitude of 0.1 m to help localise the flow of water. The active layer is set to be 0.1 m thick and the grain size distribution $F_i[1]$ in Table A1. Precipitation rates are of 1 mm/hr

or 0.1 mm/hr across the 5000 by 10000 m domain and fall velocities of $10^{-2}$ and $10^{-4}$ m/sec are tested. Microplastics are set to be only in the active layer and over a 1000 by 2000 m rectangle within the centre of the model domain (e.g. Figure 5c).

### 5.1    Microplastic transport in low flows

The overland flow of water collects within the valley as would be expected. For low precipitation rates the flow height is low and the channel width is only one cell wide (Figure 5a and b). When the fall velocity is likewise low, the result is that the

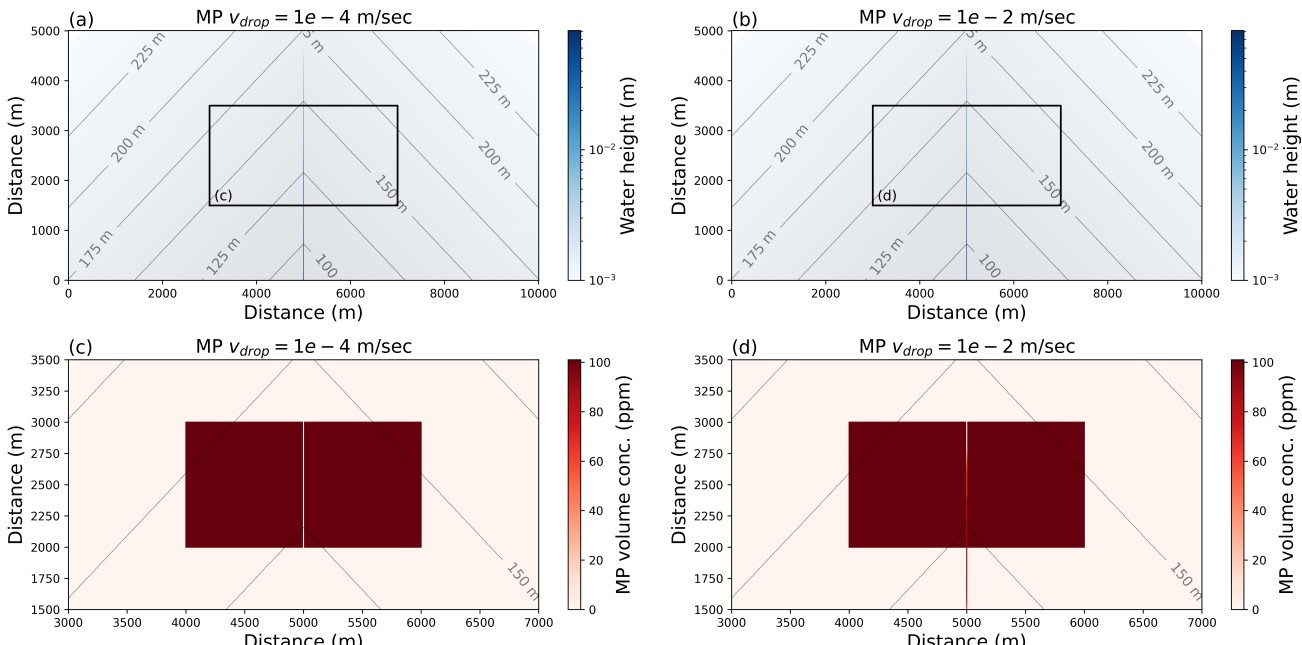

**Figure 5.** Water flow and microplastic concentration in the active layer after 60 hr when the precipitation rate is 0.1 mm/hr and the fall velocity of the microplastic is $10^{-4}$ or $10^{-2}$ m/sec (microplastic grain size of 300 $\mu$m and median sediment grain size of 110 $\mu$m). (a) Water height when the microplastic fall velocity is $10^{-4}$ m/sec in the blue color map and the topography is contoured at 25 m intervals. The black rectangle shows the bounding box for part c. (b) Water height when the microplastic fall velocity is $10^{-2}$ m/sec. The black rectangle shows the bounding box for part d. (c) Volume concentration of microplastic in the active layer when the fall velocity is $10^{-4}$ m/sec. (d) Volume concentration of microplastic in the active layer when the fall velocity is $10^{-2}$ m/sec.

microplastic is stripped in the region where the water flow crosses the rectangular source zone (Figure 5c). When the fall velocity is faster microplastic gets deposited within the channel (Figure 5d).

In the two model cases with a low precipitation rate of 0.1 mm/hr the water flow is very low, and there is limited erosion of the active layer. The erosion is so low such that the active layer is not eroded sufficiently to require the addition of new material from the strata below. The microplastic flux out from the model domain for a fall velocity of $10^{-2}$ m/sec is zero (Figure 6). When

the fall velocity is $10^{-4}$ m/sec a pulse of microplastic leaves the model domain in a time window of around 24 hrs, with a second minor increase in microplastic as the water flux increases towards the steady state and the model cells at the edge of the central channel are eroded (Figure 6).

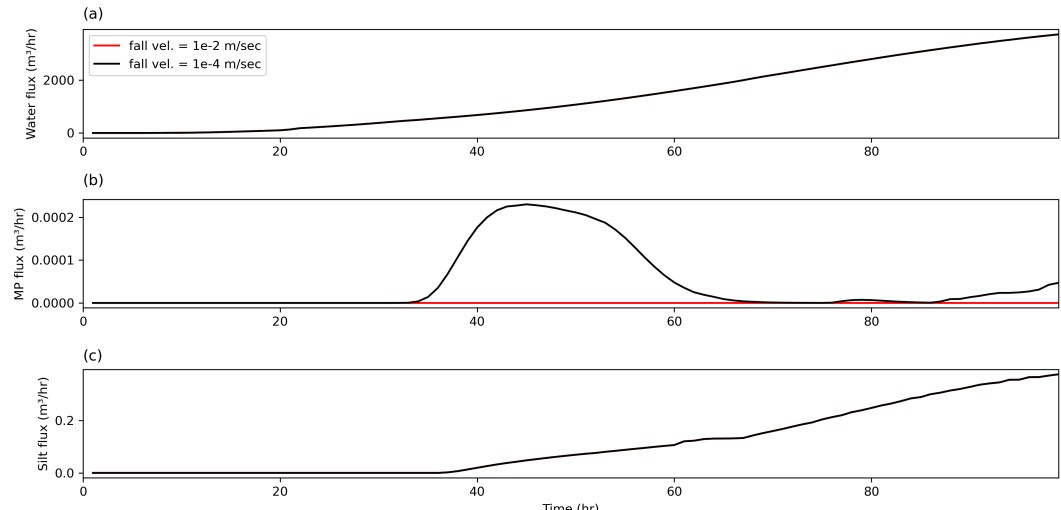

**Figure 6.** Water, microplastic and silt flux from the model domain when the precipitation rate is 0.1 mm/hr. (a) Water flux, note there is only one curve as the fall velocity of microplastic does not significantly impact the topography to alter the water flux. (b) Microplastic flux, where no microplastic leaves the model domain on the 100 hr time window if the fall velocity is $10^{-2}$ m/sec while there is a pulse of microplastic when the fall velocity is $10^{-4}$ m/sec. (c) Silt flux, where the output from the two models are identical, the microplastic does not influence suspended sediment fluxes. The fall velocity for silt is fixed at 0.00273 m/sec.

## 5.2 Microplastic transport in high flows

For the two models with a higher precipitation rate of 1 mm/yr the water depth is significantly higher, up to 0.3 m (Figure 7a and b). The higher flux of water means the channel is wider and more microplastic gets stripped from the active layer. For a high fall velocity a significant quantity of microplastic gets deposited within the channel (Figure 7d).

The increased water flux means that the active layer becomes significantly eroded, such that the strata below become incorporated within the active layer. The incorporation of the lower strata is based on the criteria that the thickness of the active layer falls below one quarter of its initial thickness then the layer below is added. This means that cells will change in thickness of active layer and grain size distribution. Therefore, if there is a slight difference in thickness in adjacent cells causing one cell to meet the criteria while the neighbouring cell does not, erosion rates will vary between the two cells due to the different relative grain size proportions. This will then cause a cascade in different local erosion rates, creating the *rough* nature of the active layer thickness of the channel bed at the end of the model run (Figure 8).

This local change in active layer thickness and local grain size distribution has the effect of creating a sediment flux output that is unsteady through time (Figure 9). The movement in the sediment flux is not due to the water routing, but due to the shifts in grain size distribution as the active layer is adjusted locally due to erosion. The occurrence of microplastic influences the silt flux within the model, as evidenced in Figure 9. If the sediment transport was not affected by microplastic, then the silt flux would be identical for both models. However, for the two models that are identical except for the microplastic fall velocity, the

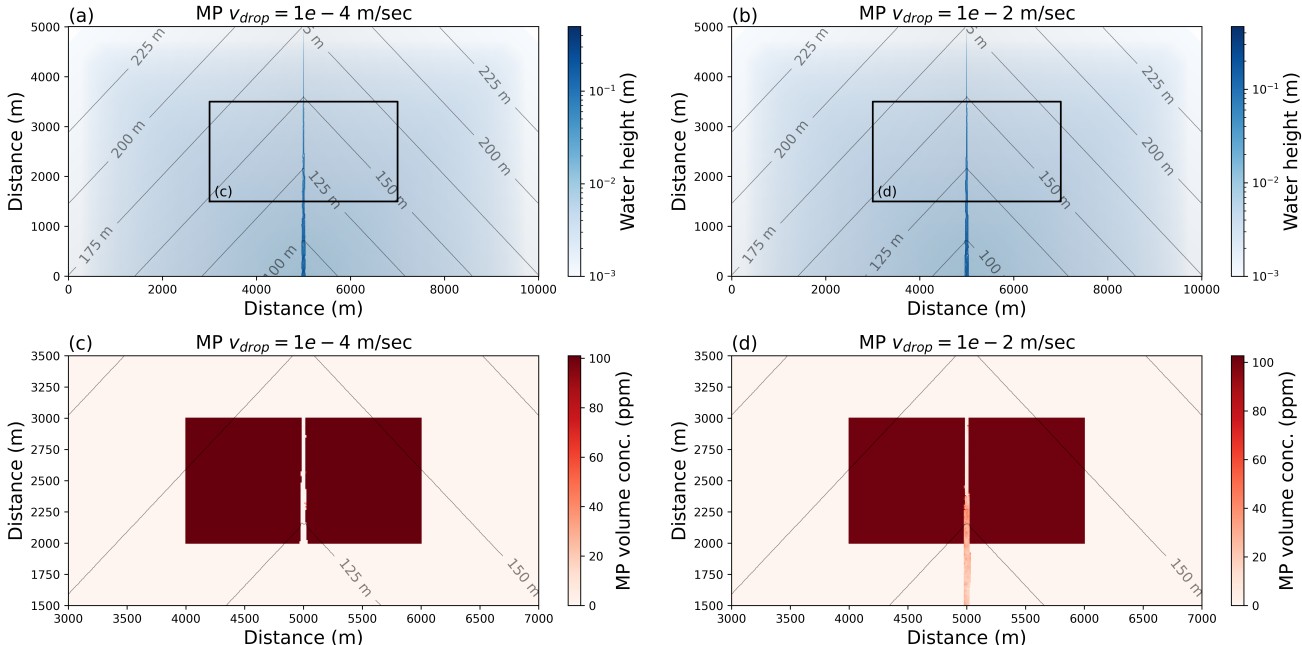

**Figure 7.** Water flow and microplastic concentration in the active layer after 60 hr when the precipitation rate is 1.0 mm/hr and the fall velocity of the microplastic is $10^{-4}$ or $10^{-2}$ m/sec (microplastic grain size of 300 $\mu$m and median sediment grain size of 110 $\mu$m). (a) Water height when the microplastic fall velocity is $10^{-4}$ m/sec in the blue color map and the topography is contoured at 25 m intervals. The black rectangle shows the bounding box for part c. (b) Water height when the microplastic fall velocity is $10^{-2}$ m/sec. The black rectangle shows the bounding box for part d. (c) Volume concentration of microplastic that remains in the active layer when the fall velocity is $10^{-4}$ m/sec. (d) Volume concentration of microplastic that remains in the active layer when the fall velocity is $10^{-2}$ m/sec.

silt flux is likewise different. The microplastic must therefore be impacting the sediment transport, and this can only be achieved within the model if the proportions of grains are different within the active layer.

Furthermore, the model would suggest that the addition of the small amount of microplastic within the source region will impact the release of silt from the model domain (Figure 9). This is due to the deposition of microplastic impacting the active layer thickness and so the outcome for the erosion of silt and the other sediment grains is not the same for the two model scenarios.

## 6  Landscape scale application - Têt catchment, France

### 6.1  Topography and hydrology

The Têt River is a typical coastal Mediterranean river with a drainage catchment area slightly less than 5000 km$^2$. The coastal plain of the Têt catchment is characterized by agricultural activities, as well as the location of the largest city of the district

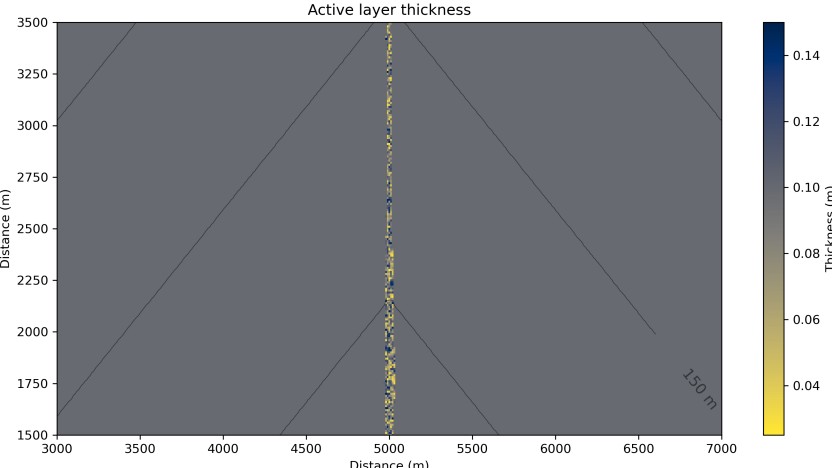

**Figure 8.** Thickness of the active layer within the microplastic source region after 60 hours of model run time for the model run where the precipitation rate is 1.0 mm/hr and the fall velocity of the microplastic is $10^{-4}$ m/sec (microplastic grain size of 300 $\mu$m and median sediment grain size of 110 $\mu$m).

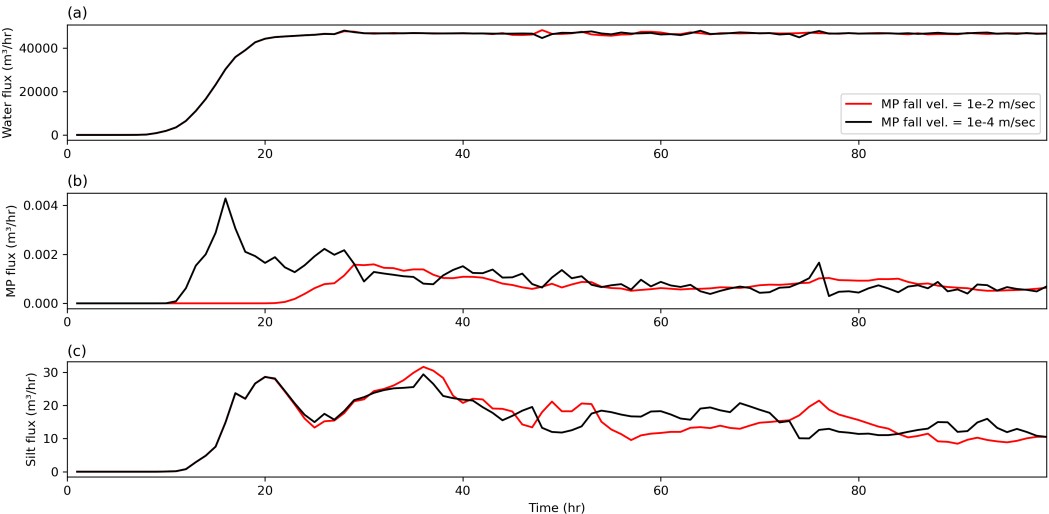

**Figure 9.** Water, microplastic and silt flux from the model domain when the precipitation rate is 0.1 mm/hr. (a) Water flux, where the water flux is not identical for the two model runs due ot the topographic change from erosion of microplastics from the bed at the higher water flow rates. (b) Microplastic flux, where both model transport the microplastic sufficiently that it starts to be exported out from the domain. When the fall velocity is slow, $10^{-4}$ m/sec, there is a pulse of microplastic export. (c) Silt flux, where the output of silt is modified by the presence of the microplastic. The fall velocity for silt is fixed at 0.00273 m/sec.

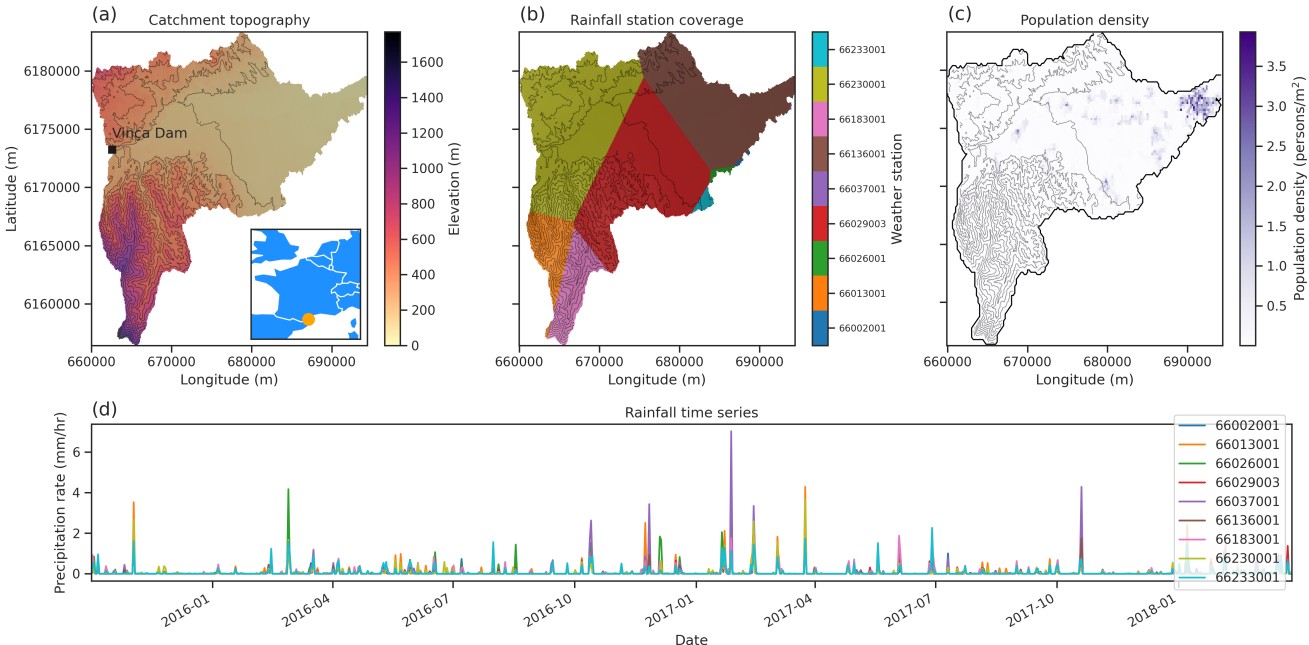

**Figure 10.** Location, topography, rainfall, and population density for the Têt catchment below the Vinça Dam. (a) Topography with location map inset where the orange marker is the location of the Têt catchment and the Vinça Dam is marked by the black square. (b) Zones used for each rainfall gauge within the model catchment, where the rainfall gauges are numbered from 66002001 to 6623001 from the identification numbers of the weather stations from Météo France (see Publithèque of Météo France). The zonation is a simple nearest neighbour algorithm using the python library `scipy`. (c) Population density of the Têt catchment below the Vinca dam at a 200 m grid resolution. The data is from the INSEE Filosofi population database. (d) Daily rainfall data time series data used to force the model for the nine weather stations numbered from 66002001 to 6623001 from the identification numbers of the weather stations from Météo France (see Publithèque of Météo France).

(Perpignan, about 150,000 inhabitants). Two major dams exist in the Têt River basin and they lie upstream the densely populated
coastal plains. The most downstream dam is at Vinça (Figure 10a). This dam is used to manage water resources during summer time and control flood events. To simplify the application of the microplastic transport model we will focus on the catchment downstream of the dam (see Figure 10a). This region has low slopes and therefore is appropriate for the diffusive-wave approach of the overland flow model. Furthermore the alluvial plain contains the source region of microplastics due to the increased quantity of roads and population centers here.

The model is forced with a daily time series of rainfall from eight weather stations that are within or near the catchment (Figure 10b and d), where the data is from the Publithèque of Météo France. The zones for which each time series is applied is calculated using the nearest neighbour algorithm from the `scipy` python library. Given the low resolution of the rainfall data we chose to keep the spatial resolution at $200 \times 200$ m per cell. Noting that `CAESAR-Lisflood` is resolution dependent (Skinner

and Coulthard, 2023), a resolution of 200 m is appropriate for testing the applicability of the simple microplastic transport model given the broad assumptions on transport and the lack of detailed rainfall data at a high spatial and temporal resolution.

To avoid artificial damming of the river due to artifacts within the digital elevation model, we pre-process the DEM. The DEM is downloaded from the BD Alti data portal of the IGN (Institut national d'information géographique et forestière). We down-sample the DEM to 200 m cells using both the mean and minimum values. Using the mean and minimum values we run first the pit-filling algorithm `SinkFiller` (Barnes et al., 2014) from the `LandLab` library (Hobley et al., 2017; Barnhart et al., 2020; Hutton et al., 2020) on both resampled DEMs, using the D4 routing to be consistent with the D4 nature of the Lisflood-FP algorithm implemented in `CAESAR-Lisflood`. Focusing on the minimum DEM we run the Lisflood-FP algorithm looped over two years of rainfall to obtain the river network. Regions where the water depth is greater than 0.05 m are then used to replace the mean value with the minimum value. Therefore, the pre-processed DEM consists of the mean value from the high resolution DEM on the hill slopes, and the minimum value within the channels.

Punctual measurement of microplastic concentrations at 10 km inland from the Têt river outlet were carried out in 2016 using nets that can capture microplastic particles larger than 330 $\mu$m (Constant et al., 2020). Combined with the gauging station there are therefore observations for water flux and microplastic. We first calibrate the hydrological model against the observations of water flux. The two key parameters to which the hydrological component of `CAESAR-Lisflood` is sensitive are the assumed quantity of rainfall that passes through evapotranspiration into run-off, and the storage of water within the subsurface via the `TOPMODEL` parameter $m$ (Remaud et al., 2024). For evapotranspiration we make the simplified assumption that it is constant in time and reduce the precipitation by using 80, 60 or 40% of observed daily rate. `TOPMODEL` is a simple set of logarithmic functions that approximate the hydrographic response to precipitation to give the characteristic recession curve within a river after precipitation input (Beven et al., 1984). The parameter $m$ controls the hydrographic response, where smaller $m$ leads to a faster return in the run-off to the steady state after a rain-fall input. We test the range of $m$ = 0.006 to 0.014.

To compare the model water flux with the observations we focus on the spring and autumn months, from 15/2/2016 to 1/5/2016 and from 1/9/2016 to 1/11/2016, where the discharge is not significantly altered by the Vinça dam. The quality of model fit is estimated using the Nash-Sutcliff model efficiency (NSE), where if the fit is negative, then the model is no better than the time averaged mean, while good fits are closer to 1 (Figure 11). We find that most models do not give a good fit, however if we assume that 40% of precipitation effectively reaches the surface, and a `TOPMODEL` $m$ parameter of 0.008 to 0.010 then the NSE value is 0.46 to 0.47. Our preferred best-fit model is highlighted by the blue box in Figure 11, where $m = 0.010$ and the effective precipitation is 40% of the observations.

### 6.2 Microplastic source and concentrations

Microplastics primarily enter river systems through waste water and the mismanagement of waste. The quantity of microplastics entering the rivers can be significantly reduced through waste water treatment, although microplastics related to fabrics are still transported into the rivers (Woodward et al., 2021). From looking at samples from riverbank deposits, it was found that in the River Tame, UK, microplastic is sourced from both treated waste water and more importantly untreated waste. Two sources are highlighted (Woodward et al., 2021): (1) continuous transport at low concentrations of synthetic fibers from treated waste water

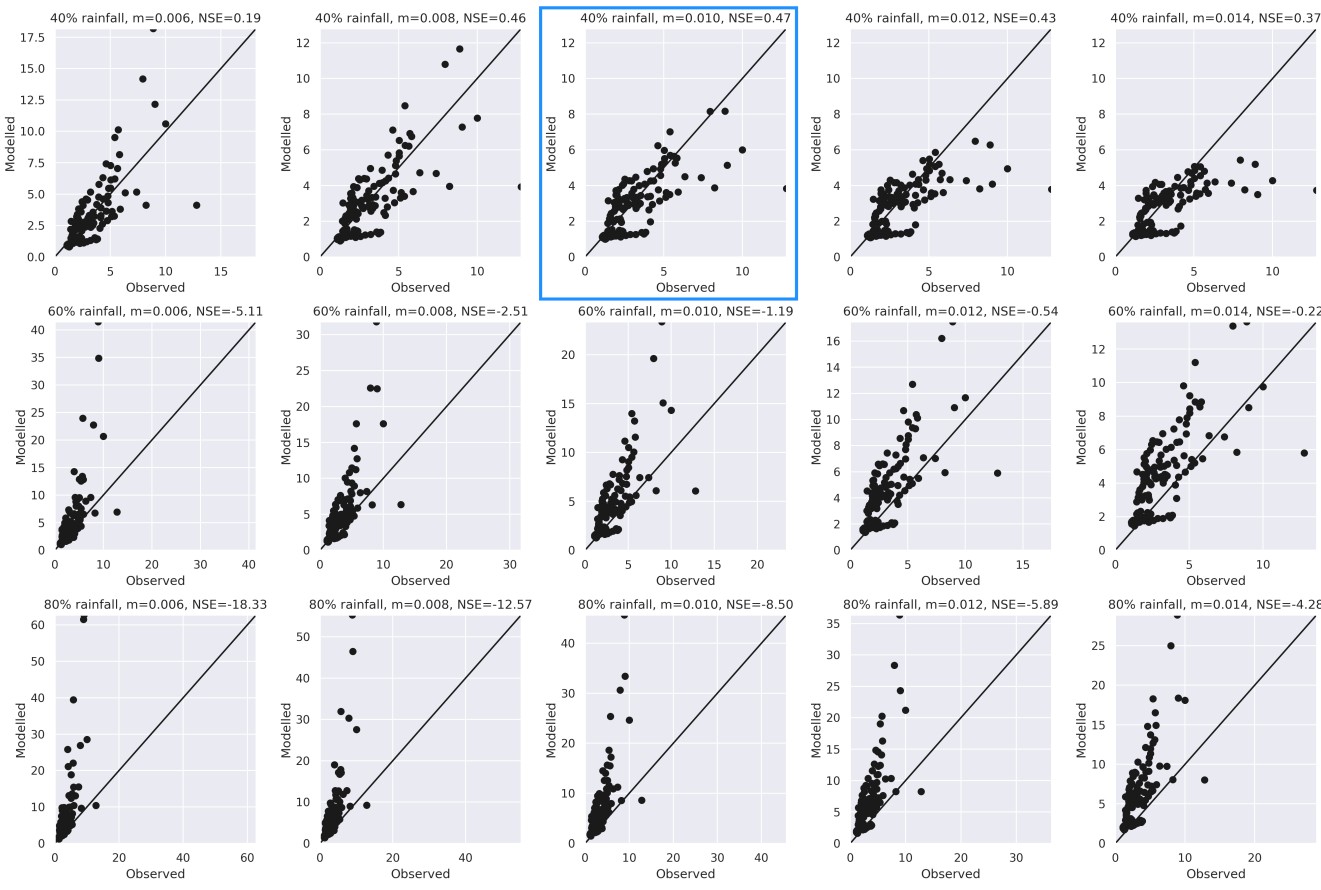

**Figure 11.** Nash-Sutcliff model efficiency (NSE) for the model for different assumptions on rainfall loss due to evapotranspiration, 40, 60 or 80% of rainfall is transmitted from the surface into the subsurface as input to the `TOPMODEL` component of `CAESAR-Lisflood` and surface storage via the `TOPMODEL` parameter $m$ that controls the peak and duration of the recession curve. If the NSE is negative the model fit is no better than the a constant mean value, the closer to 1, the better the fit.

**Table 1.** Range of tested microplastic concentrations (ppm) in active layer

| Population density (persons/m$^2$) | *high* MP concentration (ppm) | *mid* MP concentration (ppm) | *low* MP concentration (ppm) |
|---|---|---|---|
| $\geq 2$ | 100 | 10 | 1 |
| $< 2$ and $\geq 0.5$ | 10 | 1 | 0.1 |

effluent; and (2) episodic flood-driven transport of the full microplastic assemblage entrained from contaminated channel beds. The evidence from this catchment would suggest that there are point sources of continuous microplastic from urban treatment works and factories. Flood waters will however source microplastics that have become deposited within the river catchment and hence there are events where microplastic is sourced catchment wide. In a similar study on the Brisbane River, Australia, it was found that the concentration of microplastic deposited at the riverbank did not vary spatially along with land use but the types of plastic found varied (He et al., 2020). In rural areas microplastic deposition was dominated by polyethylene (He et al., 2020).

In water treatment stations, the dense microplastics are typically removed through settling and remain in the sludge. Microplastics within fabrics are significantly harder to remove, hence they seep into the rivers along with the treated water. The sludge is however processed and used in many countries as a fertiliser. This raises the potential that microplastic enters rivers from runoff withing agriculture land (Nizzetto et al., 2016), and might explain the lack of variability in microplastic concentrations in the Brisbane River. Atmospheric falls could also act as a source of microplastic in soil and along catchment slopes (Zhang et al., 2020). Regardless, microplastics are ubiquitous within the environment, and with the current lack of continuous monitoring within any fluvial system, current estimates on the sources of microplastic are extrapolated from only a handful of measurements.

To estimate the source of microplastics within the catchment we make the assumption that microplastics of 300 $\mu$m in size in the active layer of the model are related to the population density (Weiss et al., 2021). Using the INSEE Filosofi population database averaged of a 200 m raster (INSEE and Ministère des Finances (DGFiP), 2017), we created a map of the population density (Figure 10c). We then relate the population density to the volume concentration of microplastic within the active layer in the catchment, where if the the population density is greater than 2 persons per square meter then the volume concentration of microplastics are high, while if the population density is between 0.5 and 2 persons per square meter the concentration is low, and below this population density we assume zero microplastic pollution (see Table 1). There is very little information on the quantity of microplastic in surface soils within rural France (e.g. Kedzierski et al., 2023). Mass concentrations of microplastics observed in soil samples globally show a large variability (Rohais et al., 2024). From a total of 107 measurements the median soil mass concentration of microplastic is 44 mg/kg, with first and third quartiles at 1.50 mg/kg and 674 mg/kg (Rohais et al., 2024). In relatively polluted soil, such as dredge samples of the Aa River in northern France that is in a industrial area, there is for example up to 100 mg/kg of microplastic (Constant et al., 2021). Based on these measurements we propose three catagories of *high*, *mid* and *low* volume concentrations of the microplastic (see Table 1).

The model is set up with an active layer that has a thickness of 0.5 m and it is only the initial active layer that contains microplastics within the regions defined by their population density (Figure 10c and Table 1). The sediment grain size distribution

is weighted towards silts and very fine sands (distribution $F_i[1]$ in Table A1). We run the model over the duration of the rainfall series from October 2015 to April 2018 (Figure 10d) to wind up the model and then compare to the observations in 2018. The punctual observations of microplastic concentrations can be used to verify if the model is capable of generating sensible values

for the quantity of microplastic in transport. We explore both the range of concentrations of microplastic in the active layer related to the population density (Table 1) and three fall velocities, $10^{-2}$, $10^{-4}$ and $10^{-6}$ m/sec. Given the observations are punctual and do not capture peak flow conditions we will not look at the time series, but instead look at the relationships modelled and observed between water flux and microplastic concentrations (Figure 12).

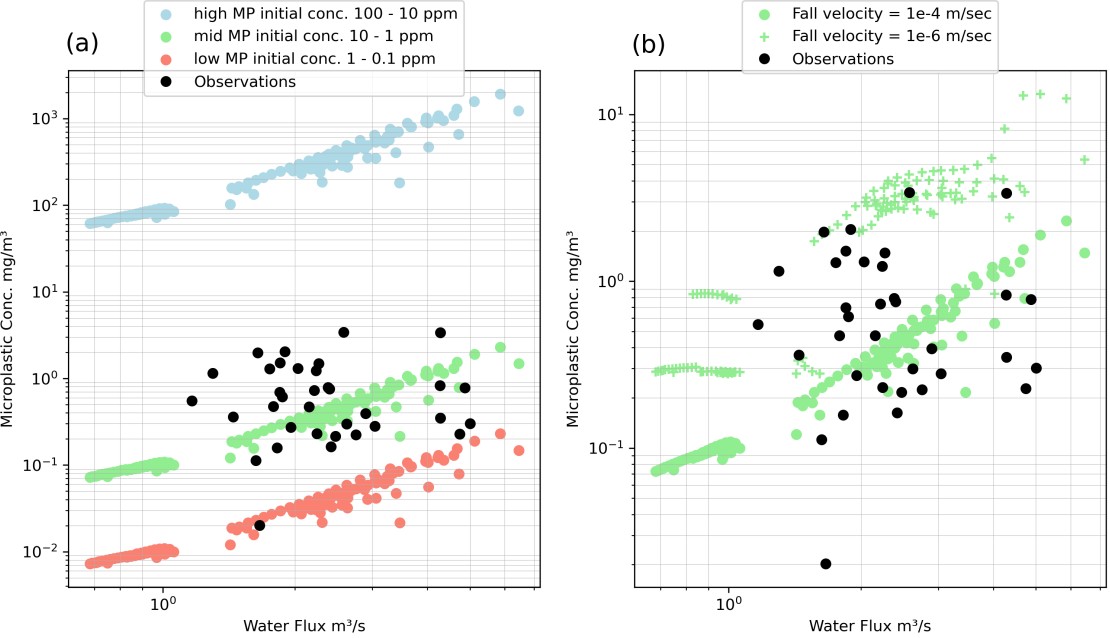

**Figure 12.** Comparison of observed and modelled microplastic concentrations at the outlet of the Têt catchment (Constant et al., 2020). (a) Different initial concentrations of microplastics (Table 1) assuming a fall velocity of $10^{-4}$ m/sec. The volume concentration is converted to a mass per volume of water assuming the microplastic density is 1300 kg/m$^3$. (b) Different fall velocities assuming initial concentrations of microplastics that is in the *mid* range (10 - 1 ppm).

From a comparison of the model microplastic fluxes and the observations it is clear that the *high* initial contamination of

microplastics generates more microplastic in suspension than the observations (Figure 12a, light blue). The *low* contamination model generates too little microplastic (Figure 12a, salmon red), while the *mid* contamination generates a trend in microplastic concentrations that is coincident with the observations (Figure 12a, light green). This implies that under the assumptions of this model and at a resolution of 200 m cells, the microplastic volume concentrations related to populated areas could be in the range of 1 to 10 ppm. The fall velocity of the microplastics will however impact the flux out of the system. For a fall velocity of

$10^{-2}$ m/sec no microplastic is exported from the basin in the 5 years of precipitation modelled. For a fall velocity of $10^{-4}$ the

microplastic concentration within the water column is within the range of the observations (Figure 12b). If the fall velocity were at the extreme low end of observations for silts, $10^{-6}$ m/sec (Dietrich, 1982), then the model concentrations are at the upper limit of the observed microplastic concentrations (Figure 12b).

If we assume that the *mid* model of initial microplastic contamination and a fall velocity of $10^{-4}$ m/sec is representative of the Têt catchment discounting the area above the Vinça dam, then microplastic becomes entrained and deposited within the Têt fluvial system (Figure 13). As with the tests on the rectangular catchment (Figure 7 and Figure 8), we can see that the model creates a channel with a distribution of regions of low and high active layer thickness as this layer becomes eroded and is unified with the strata below (Figure 13a). The water height within the alluvial plain is of the order of meters, which is consistent with the Têt river, given the low resolution of the model (Figure 13b). Within this channel, after a model evolution of 5 years the microplastic has started to accumulate within reaches of the Têt, particularly to the west of the more populated region of Perpignan and at the outlet (Figure 13c).

## 7  Discussion

In this paper we have outlined a relatively simple method for including microplastic transport within a landscape evolution model, where the overland flow of water is solved and a shear stress on the bed can be approximated. The motivation for developing this model was to explore the transport and storage of microplastic within fluvial systems. Studies that have looked at the storage of microplastic within rivers have found locally very large accumulations (Hurley et al., 2018), and it has also been noted that microplastic accumulation can be associated with the accumulation of silt (He et al., 2020). We therefore decided to explore if we could apply simple rules developed for sediment transport to capture the transport of microplastic. The model developed here clearly has its limitations, which we will outline later. First we will explore what the model implies for microplastic contamination.

Retention of microplastic is a function of the fall velocity. For the simple rectangular catchment there was minimal retention of microplastic when the fall velocity was $10^{-4}$ m/sec for both low and high water flux (Figure 6 and Figure 9). This can be seen in the small flux of microplastic that continues to be released after the main pulse of microplastic. When the fall velocity was two orders of magnitude higher, $10^{-2}$ m/sec, there was significant retention of microplastic. However, when the model was scaled to the Têt catchment we find that a fall velocity of around $10^{-4}$ m/sec best matches the observed concentration of microplastics within the water (Figure 12b). In settling columns, the fall velocity of microplastic particles are of the order of $10^{-3}$ to $10^{-1}$ m/sec (Waldschläger and Schüttrumpf, 2019b; Constant et al., 2023), which is similar to fine sand and soil (Constant et al., 2023). Settling columns ignore processes such as saltation, that might keep more of the microplastic in suspension rather than in a cycle of deposition and erosion. This might explain why at the landscape scale the model implies lower fall velocity. Otherwise, the model developed here would suggest that a significant quantity of microplastic will be stored within the fluvial network.

The model also implies that there is of the order of 1 to 10 ppm volume concentration of microplastic within the active layer at a $200 \times 200$ m resolution (Figure 13a). A correlation between population density and the generation of microplastic within mismanaged waste has been recognised, however the estimates are reported as a mass concentration (Weiss et al., 2021). The

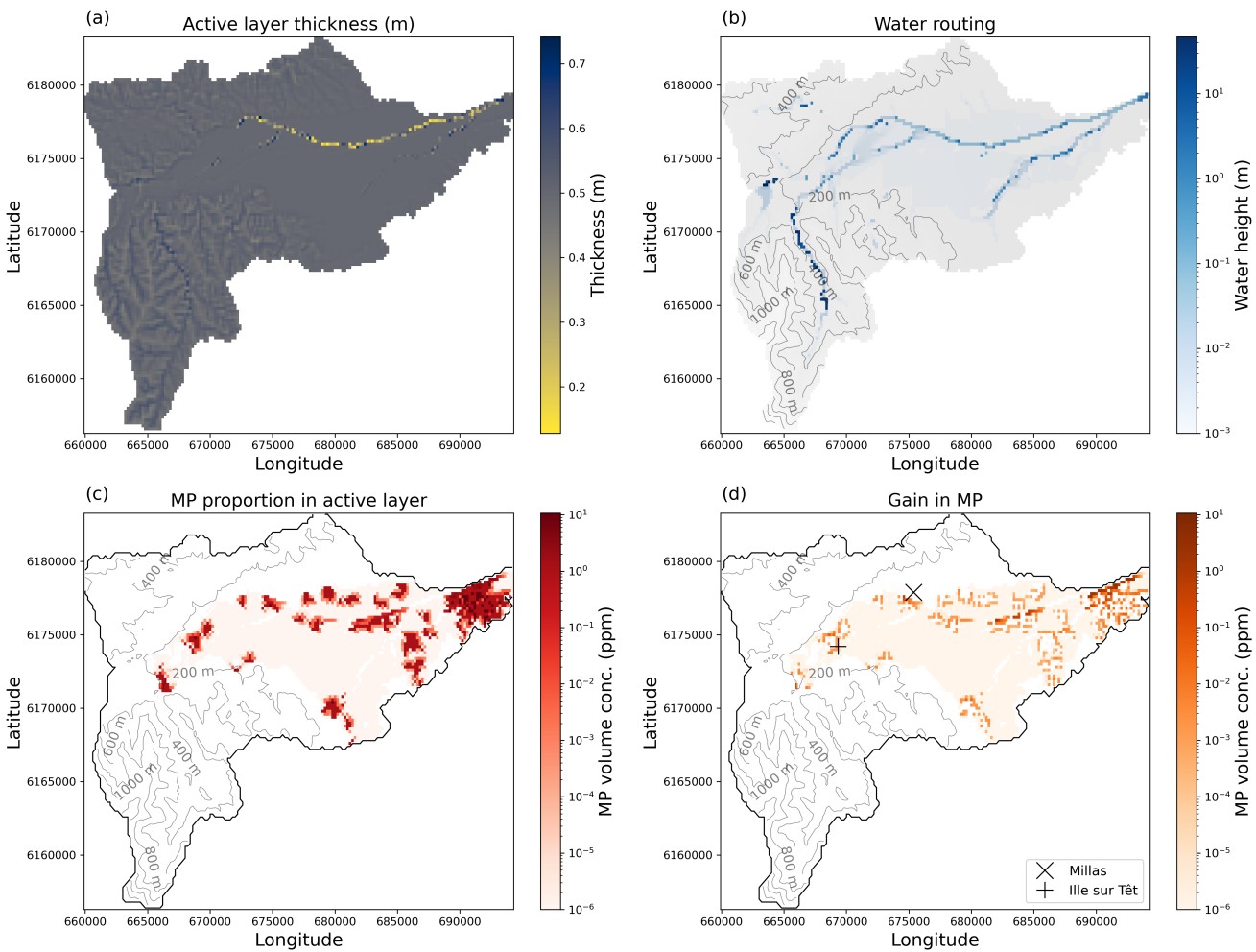

**Figure 13.** Change in the active layer, water depth, and concentration of microplastics for the *best fit* model scenario. (a) Thickness of the active layer after the 5 year model run time, where the initial thickness was 0.5 m. (b) Water height within the model domain. (c) Volume concentration within the active layer as a function of the initial condition and the transport and deposition of the microplastic after 5 years of daily rainfall history. (d) Positive change in the microplastic concentration. The microplastic is observed to become deposited along the river network below each small populated zone.

difficulty is in converting this to a volume concentration as a model input, due to the large range of possible soil densities. In this study, by finding the model source concentrations that fit the observed suspended microplastic load, we can estimate the distribution of microplastic within the landscape. Making the assumption that populated areas lead to microplastic pollution, we find that reaches of the river Têt will become contaminated with microplastics down-system of these source zones. For example, down stream of Ille-sur-Têt and Millas, the model suggest that microplastic will be stored within the river sediments (Figure 13d). From a compilation of river sediment observations (Eo et al., 2019; Kabir et al., 2022; He et al., 2020; Klein et al., 2015; Niu et al., 2021; Rao et al., 2019; Rodrigues et al., 2018; Sarkar et al., 2019), the observations would suggest that there is on average the order of 50 mg/kg of microplastic within the river sediments (see Rohais et al. (2024) for the full compilation). If we assume that the density difference between the microplastic and sediment is roughly a half and ignore the porosity then the volume concentration is of the order of 25 ppm. This is in the range of the model values and redistribution of microplastic within the model (Figure 13c). It would be ideal if in future work the sediment could be sampled in these regions to obtain an idea of the concentration of microplastic.

Microplastic is included within the model only as part of the surface active layer. As the active layer is eroded below the threshold thickness there is an instantaneous reduction in the volume concentration of microplastic within the updated active layer, accompanied by a change in the transport properties of the active layer within the cell. For the 2D model, the initial condition is of a rough surface generated by the addition of an array of random fractions added to the smooth topography. This roughness causes the flow to increase non-uniformly and generate a distribution of cells that become eroded and modified that is a function of the seed used to generate the array of random fractions (Figure 8). The impact of active layer on fluvial erosion is not the focus of this paper, however for the simple 2D case, there is evidence that the active layer is eroded as a wave of incision migrating up the catchment. However, when the model is applied to the Têt catchment, the resolution becomes too low to make any meaningful interpretations on the patterns of erosion within the river channel. The overall increase in microplastic concentration downstream in the river, nevertheless, suggests that microplastic pollution is accumulating as water flows downstream and could deserve special attention for monitoring and remediation.

The model does however have a few limitations that are worth discussing. The first is related to the diffusive-wave approximation to solving the shallow water equations for overland flow. In order to have stable solutions it is necessary to limit the water flux such that the Froude number for the flow is less than 1. This means that when the slope exceeds 2° the water velocity does not increase with increasing slope. As such the transport model will underestimate the quantity of sediment and microplastic eroded within steep topography. Second, simulations with `CAESAR-Lisflood` are known to be resolution dependent (Skinner and Coulthard, 2023). The resolution dependence on flow routing is a known problem in landscape evolution models, which could potentially be overcome with filtering techniques (Coatléven and Chauveau, 2024), however this is beyond the scope of this paper. It is worth noting that while the 1D tests of the numerical implementation of the microplastic transport suggest that above a certain resolution the model is stable, this might not be the case for the 2D runs. This touches a related problem of keeping the model resolution coherent with the resolution of the observations. For the test against the Têt catchment we have kept the resolution low, as we have daily rainfall data at point locations, and limited observations of microplastic flux. It is difficult therefore to argue for a increased resolution for the model. Finally, `CAESAR-Lisflood`, as with all process-based

models, is sensitive to the model parameter choices (Skinner et al., 2018). While we have calibrated the hydrological model to
the gauge station data, the sediment and microplastic flux is strongly sensitive to the Mannings roughness coefficient and will
also be strongly dependent on the choice of topographic slope at the outlet of the catchment and assumed grain size distribution
(Skinner et al., 2018; Remaud et al., 2024).

## 8 Conclusions

In sedimentology rivers are the pathway for sediment from source-to-sink and as such the transport of sediment by flowing water
has been given a great deal of attention. Landscape evolution models have since been developed with the aim of capturing the
basic principles of the transport of sediment to capture the evolution of fluvial systems. Microplastic form dense particles within
the water column that may remain in suspension for a considerable amount of time, equivalent to suspended sediments, or may
settle and become trapped within fluvial deposits. From this descriptive similarity we have developed a simple set of laws to
capture the transport of microplastic, the erosion of microplastic from the river bed, and the deposition of microplastic. This
allows us to insert microplastic into a landscape evolution model such as `CAESAR-Lisflood` and then track the deposition
of the microplastic within the fluvial network. By applying this model to the Têt catchment we find that there is potentially
high quantities of microplastic within the fluvial deposits upstream of Perpignan. By calibrating the model against observed
microplastic fluxes, we can estimate the volume concentration of microplastic within the source regions, assuming microplastic
is correlated with population density. The application of reduced complexity models to address the interaction between natural
grains and microplastics in river systems provides a promising avenue for advancing our understanding of both microplastics
and sediment transport-export dynamics.

*Code availability.* The 1D Microplastic transport model is available here:https://ifpen-gitlab.appcollaboratif.fr/1d-microplastic-fluvial-transport/microplastic1d.

The code for `CAESAR-Lisflood` (`HAIL-CAESAR`) with the modifications is available here: https://github.com/johnjarmitage/HAIL-CAESAR/tree/plastic.

Notebooks to plot the output of the model runs are available here: https://github.com/johnjarmitage/caesarPy/tree/master/notebooks/tet.

The workflow for the DEM pre-processing is available here: https://github.com/johnjarmitage/caesarPy/tree/master/dem-preprocessing

*Author contributions.* Both authors contributed to the study conception and design. Methods and model development were performed by John
Armitage. Analysis and interpretation was performed by John Armitage and Sébastien Rohais. The first draft of the manuscript was written by
John Armitage and both authors commented on previous versions of the manuscript. Both authors read and approved the final manuscript.

*Competing interests.* The authors have no relevant financial or non-financial interests to disclose.

*Acknowledgements.* We would like to acknowledge Wolfgang Ludwig and Mel Constant for sharing their data and discussion on microplastic, and Declan Valters and Tom Coulthard for sharing their code and time in explaining various input variables and parameters.

**Table A1.** Grain size distributions

| grain | $F_i[1]$ | $F_i[2]$ | $F_i[3]$ | $F_i[4]$ | $F_i[5]$ |
|---|---|---|---|---|---|
| $D_m$ (mm) | 0.11 | 0.28 | 0.56 | 1.13 | 3.87 |
| microplastic | 0.0001 | 0.0001 | 0.0001 | 0.0001 | 0.0001 |
| silt | 0.3244 | 0.0930 | 0.0265 | 0.0054 | 0.0004 |
| very fine sand | 0.2335 | 0.1423 | 0.0646 | 0.0204 | 0.0025 |
| fine sand | 0.2206 | 0.2248 | 0.1469 | 0.0667 | 0.0135 |
| medium sand | 0.1375 | 0.2352 | 0.2215 | 0.1450 | 0.0474 |
| coarse sand | 0.0608 | 0.1738 | 0.2353 | 0.2214 | 0.1170 |
| granule | 0.0187 | 0.0894 | 0.1740 | 0.2354 | 0.2009 |
| pebble | 0.0047 | 0.0415 | 0.1312 | 0.3055 | 0.6162 |

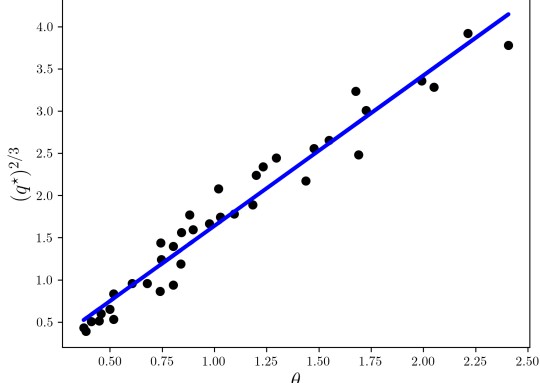

**Figure A1.** Relationship between the dimensionles flux $q^\star$ and Shields number $\theta$ for the laboratory experiments of the transport of plastic beads down an inclined slope (Berzi and Fraccarollo, 2013). The gradient of the linear regresion is 1.92 between $(q^\star)^{2/3}$ and $\theta$, suggesting $C_p \approx 2.4$ in equation (11).

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
