# Peer review of "A numerical model of microplastic erosion, transport, and deposition for fluvial systems"

_EGUsphere, 2024_

## Author Response (AR1)

We would like to thank the two reviewers for their constructive and useful questions and comments. Below we reply to the major comments in the order that the reviewers listed them.

Reviewer comments are in plain text.

Author replies are in bold text.

Reply to reviewer 1

RC-1: Sediment transport is calculated from Wilcock and Crowe (2003), but microplastic transport is calculated from Meyer-Peter-Muller (ln 125). Why the difference? Why not use Willcock-Crowe for microplastics as well? Please explain. Also, why use Meyer-Peter-Muller specifically for the microplastics? Why not any particular other one?

**This was a pragmatic choice rather than a choice driven by a preference of one model over another. There exist many different empirical and heuristic models for sediment transport, of which two are implemented within the C++ version of CAESAR-Lisflood (Einstein's model and Wilcock and Crowe,2003). In a review of these different models, Parker (2008), https://doi.org/10.1061/9780784408148.ch03, it was suggested that the Einstien model is not adequate for describing the motion of sediments. Therefore, we used the Wilcock and Crowe (2003) model and for consistency implemented it within the simplified 1D code. Subsequently for the microplastic we chose to use the Meyer-Peter-Muller approximation as this lends itself more naturally to the notions of a critical shear stress for which there is some preliminary literature in the case of microplastics. We do not have the same range of experiments to find the equivalent parameters for the Wilcock and Crowe (2003) parameterisation. Furthermore, it is worth noting that the Wilcock and Crowe (2003) model is based on experiments on sands, and therefore extrapolation down to smaller particle sizes might be dangerous.**

**Added text:**

**(line 130 of track changes file)**

**To estimate the flow of sediment from the flow of water there are a few empirical flow calculations that can be used, see Parker (2008) for a review. In the landscape evolution model CAESAR-Lisflood, the multi-grain model of Wilcock and Crowe (2003) is implemented. This model has the advantage that it can approximate the transport properties of multiple grain size fractions. It is however based on an extrapolation from laboratory experiments, and it is not straightforward to include microplastics as within this model. We therefore chose to include microplastics separately using a Meyer - Peter and Müller law as the hiding function have already been estimated for microplastics.**

RC-2. The model uses a single microplastics size (ln 130). As microplastics range from approximately 1 μm to 5000 μm (Weber et al., 2022, Sci Tot Env, 819), this covers a wide range of sizes. Is it realistic to represent this variation as one size in the model?

**It depends on the focus of the study. Given the number of approximations, the lack of observations, and the increase in computation cost with each additional particle modelled in suspension, we are not comfortable with introducing a range of microplastic sizes when we cannot be sure if the model output is useful. As for sediments, there is a good argument for sub-dividing microplastics into smaller classes, yet without the observations for how these sub-classes are**

**deposited, we think the first approach would be to keep the model complexity low and explore the fate of a characteristic microplastic size.**

RC-3. Also, in the microplastic size sensitivity analysis, the range of sizes cover essentially one order of magnitude (from 1000 μm to 1000 μm), whereas the full range spans three orders of magnitude (from 1 μm to 5000 μm). Sensitivity of fall velocities is explored over multiple orders of magnitude. So why not do the same for microplastic sizes? Is it possible that flux significantly increases as microplastics get up to two orders magnitude of smaller (cf. Figure 3b)?

**This is a valid point. When we revise the manuscript, we will expand the sensitivity analysis to microplastic (MP) grain size to increase the range from currently 100 to 1000 μm to 1 μm to 5000 μm. Additionally, the data to test/constrain the model are from net (mesh 300μm), so that the concentrations proposed integrate the entire range from 300 to 5000 μm. It supports our choice, as a first tentative modelling study, to use only 1 class of MPs in our model:**

**In the manuscript we have increased the range of MP size (Figure 3b) and the results do not change.**

RC-4. On what basis are the active layer thicknesses chosen? Active layer thicknesses vary significantly between different experiments. The active layer in the 1D model is assumed to be 1 m thick (ln 198). This seems to be an especially large value. Why such a large value? The active layer in the 2D CAESAR-Lisflood simulations is set to 0.1 m thickness (ln 250), which seems much more reasonable. The active layer thickness in the landscape-scale simulations (Têt catchment) is set at 0.5 m (ln 356), which again seems quite large.

The idea of the active layer is this is where the sediment exchange between river bed and water column occurs. So, setting this to a relative small value (e.g. 0.1 m) seems to make sense. If the microplastics are likely to occur to greater depth, maybe it would be possible to also include microplastics in the first stratum below the active layer – which presumably could be thicker as well.

**The choice of active layer thickness is a function of the purpose of the model. This was not explained sufficiently in the manuscript. For the 1D models we do not want the supply of microplastics to be exhausted before the steady state water flux is achieved. This required the active layer to be thick, 1 m. For the simple 2D models, we instead wanted the active layer to be exhausted, so that we could see a pulse of MP travel out of the domain, hence a thickness of 0.1 m. When we applied the model to the Têt catchment, the domain is discretized on a 200x200 m DEM. The active layer therefore encompasses agricultural, urban and industrial land, hillslopes, and ridges. We did not want to limit MPs to the top 0.1 meters, so the active layer was increased to 0.5 m.**

**At the reviewer states the "active layer" in geomorphology is the layer of exchange in the riverbed, however at the scale of the catchment no point in the grid is exclusively capturing the river bed, but is an approximation of different land types. We are not convinced that there is much to be gained from the effort of re-coding CAESAR-Lisflood to make the suggested adjustment.**

**New text:**

**(line 203 of track changes file)**

**The thickness of the active layer can be somewhat arbitrary (Parker 2008). Within the river bed it can be taken to be thin, and approximated from the largest grain size in transport. In a catchment scale landscape evolution model, where the model resolution is of grid sizes of 200 x 200 m, then**

the thickness of the active layer needs to encompass natural, agricultural, rural and urban land-use, and ridge tops, hillslopes and valley floors. In this study we vary the active layer thickness depending on the application of the model. In the subsequent sections we will first explore a 1D model to look at the steady state variability in microplastic transport, then we test a simple 2D setup, and finally a catchment scale test (Figure 2). For 1D model scenario tests, we define a thick active layer of 1 m so that microplastic supply is not exhausted with the aim of understanding the steady state behaviour (Figure 2a). For 2D model scenarios we keep the active layer thin, 0.1 m, to understand the transient behaviour of the model (Figure 2b). For the application to the Têt River system the model resolution is wider than an individual river channel, at 200 x 200 m. We therefore chose to set the active layer at 0.5 m (Figure 2c).

RC-5. The model's flow solver is adjusted to resolve fluxes at the cell interfaces rather than within the cells (ln 244). It is not explained why this is necessary and, more importantly, it is not explained how this is done. As this seems a rather fundamental change to one of the model's core underlying algorithms, it would be good if the authors can elaborate on this change – maybe in Supplementary Material, as to not detract from the flow of the paper. Moreover, the authors note that their change to flow algorithm has an impact on the sensitivity to spatial resolution (ln 246), but do not clarify if their new flow model is more sensitive or less sensitive to spatial resolution than the original code.

**The 1D model uses a finite volume approach to solve the shallow water equations while CAESAR-Lisflood uses a finite difference numerical algorithm, Lisflood-FP. For the 1D model the finite volume approach was taken as it has known methods to generate stable solutions to for the shallow water equations. It is beyond the scope of this work to re-write the flow routing algorithm in CAESAR-Lisflood to use a finite volume approach.**

**In the manuscript we perhaps were not sufficiently clear in how we explained the sensitivity to spatial resolution. Our 1D model converges towards the same result as resolution is increased, however it might not be the same if we developed a 2D version (see Coatlèven and Chaveau, 2024, https://doi.org/10.5194/esurf-12-995-2024, for a discussion on resolution sensitivity). When we revise the manuscript, we propose to take care to separate the 1D model resolution dependence and the 2D model resolution dependence and drop wording that might give the impression that the finite volume approach would solve resolution issues.**

**Modified text:**

**(line 474 of track changes file)**

**Second, simulations with CAESAR-Lisflood are known to be resolution dependent (Skinner and Coulthard 2023). The resolution dependence on flow routing is a known problem in landscape evolution models, which could potentially be overcome with filtering techniques (Coatléven and Chauveau 2024), however this is beyond the scope of this paper. It is worth noting that while the 1D tests of the numerical implementation of the microplastic transport suggest that above a certain resolution the model is stable, this might not be the case for the 2D runs.**

RC-6. For the landscape scale scenarios, the simulations focus on the area downstream of a dam. Is CAESAR-Lisflood then run in reach mode, with a dam release discharge in conjunction with rainfall? Or is it run in catchment mode (i.e. rainfall only), thereby ignoring any dam release flow.

**CAESAR-Lisflood is in catchment mode, and we are ignoring dam release flow. We focused calibration on time periods where rainfall and river discharge were correlated, assuming that the**

**dam was not discharging during these seasons; that is consistent with historical reporting from the Dam's management.**

RC-7. The authors note that the CAESAR-Lisflood model is sensitive to two key parameters: the assumed quantity of rainfall that passes through the evapotranspiration into run-off, and the storage of water within the subsurface. Can the authors provide a reference for this statement? And, although the model might indeed be sensitive to these two parameters, can the authors confirm that it not sensitive to other parameters.

**We explored this sensitivity in the publication Remaud et al. (2024), https://doi.org/10.57035/journals/sdk.2024.e22.1538. Sensitivity analysis of CEASAR-Lisflood has found that it is also primarily sensitive to the Manning's roughness parameter, and various numerical choices. We propose to improve the discussion of the sensitivity of CAESAR-Lisflood.**

**Modifications: Added citation in line 347 of track changes file**

**Added text to end of discussion: (line 481 of track changes file)**

**Finally, CAESAR-Lisflood, as with all process-based models, is sensitive to the model parameter choices (Skinner et al., 2018). While we have calibrated the hydrological model to the gauge station data, the sediment and microplastic flux is strongly sensitive to the Mannings roughness coefficient and will also be strongly dependent on the choice of topographic slope at the outlet of the catchment and assumed grain size distribution (Skinner et al., 2018, Remaud et al., 2024).**

RC-8. In the landscape scale scenarios, microplastic concentration is related to population in a discretized manner (ln 345-348; Table 2). Why such a discretized approach? Why not proportionally scale microplastic concentration to population with a constant factor?

**We could test a linear corelation to population density, however the uncertainty in any model is hard to quantify, so we chose to make a simple categorical approach.**

RC-9. Figure 5b: A roughly 24-hour pulse of microplastics is observed between 38 and 62 hours into the simulation. This is briefly described in the main text (ln 262). However, the model also produces an increase in microplastic flux after 90 hours into the simulation. What is the cause for this rather unexpected increase in microplastic flux?

**This slight increase is from erosion of the sides of the channel, bringing in a second minor flux of microplastic.**

**Text modified:**

**(line 290 of track changes file)**

**When the fall velocity is $10^{-4}$ m/sec a pulse of microplastic leaves the model domain in a time window of around 24 hrs, with a second minor increase in microplastic as the water flux increases towards the steady state and the model cells at the edge of the central channel are eroded (Figure 6).**

RC-10. Initial MP volume concentration for 2D test simulations is 100 ppm (Figure 6c, d). It then seems that erosion of microplastics from the active layer should have minimal impact on active layer thickness. Conversely, deposition of microplastics also does not seem to exceed 100pm (Figure 6d), so presumably should not significantly affect the active layer thickness. Yet the authors note that the unsteady sediment flux "is due to the deposition of microplastic impacting the active layer thickness" (ln 278). This raises two questions: 1) How much of the mobilized microplastic is deposited in the

thalweg? And 2) How do the authors know that it is the deposition of microplastics that is impacting the active layer thickness rather than the deposition of sediments?

**Our explanation was poorly worded. The microplastic influences the silt flux within the model, as evidenced in Figure 8. If the sediment transport was not affected by microplastic, then the silt flux would be identical for both models. However, for the two models that are identical except for the MP fall velocity, the silt flux is likewise different. The MP must therefore be impacting the sediment transport, and this can only be achieved within the model if the proportions of grains is different within the active layer.**

**Modified text:**

**(line 301 of track changes file)**

**This local change in active layer thickness and local grain size distribution has the effect of creating a sediment flux output that is unsteady through time (Figure 9). The movement in the sediment flux is not due to the water routing, but due to the shifts in grain size distribution as the active layer is adjusted locally due to erosion. The occurrence of microplastic influences the silt flux within the model, as evidenced in Figure 9. If the sediment transport was not affected by microplastic, then the silt flux would be identical for both models. However, for the two models that are identical except for the microplastic fall velocity, the silt flux is likewise different. The microplastic must therefore be impacting the sediment transport, and this can only be achieved within the model if the proportions of grains are different within the active layer.**

RC-11. The observed microplastic concentration in the Tet river catchment is largely independent of the water flux (Figure 11). Why would that be?

**The microplastic measurements are punctual and do not capture the most extreme events. Therefore, the lack of a trend might be due to the irregular sampling during data acquisition.**

Minor comments:

I do not understand the contrast set-up in this statement "it has been observed that the quantity of microplastic that enters the rivers is related to the population density, yet the focus has been on estimates for the flux of microplastic as suspended load". Why is the notion of estimating microplastic flux as suspended load problematic in the context of the microplastic amounts being related to population density? Moreover, if there is anything problematic about this notion, then that problem remains in your study as well, since your model also treats microplastic flux as suspended load.

**We agree that our phrase as written does not make sense. We wanted to note that current models that relate plastic fluxes to population are at the global scale and treat catchments as single units where the discharge of plastic is fit by a regression to the catchment populations, water discharge, relief etc. This argument will be improved.**

**Modified text:**

**(line 38 of track changes file)**

**On a global scale it has been observed that the quantity of microplastic that enters the rivers is related to the population density (Weiss et al., 2021). However, global studies treat river catchments as single point discharge points and do not include permanent and temporary storage of microplastic within the catchment.**

ln 50: Add comma after "In effect". **Done**

ln 77: "These models use the empirical transport equations for sediments developed by (Wilcock and Crowe, 2003) to link the water flux to sediment flux". Surely not all reduced complexity models use the Wilcock and Crowe equations. **Replaced with "such as"**

ln 79: Delete "CAESAR-Lisflood".  This mention is not applicable when still discussing reduced complexity models in general. **Deleted**

ln 89: Add comma after "That is". **Done**

ln 171: "The finest grain size is treated as a suspended particle". This is not necessarily true. In CAESAR-Lisflood, the finest sediment can be treated as suspended material but does not need to be. **OK, but in our case it is.**

ln 213: "At steady state the water depth …". Does the 1D model ever reach steady state? There could be a steady water flux (ln 217), but I presume the 1D model will never have a steady sediment or microplastics flux. As long as there is flow, there will be erosion, and with continued erosion, the slope would be ever reducing. Thus, true steady state would only occur if there is no further erosion (but then there also would not be any microplastic transport), or if there is a steady uplift to compensate for the erosion (but this is not mentioned).

**Modified text:**

**(line 244 of track changes file)**

**Sediment and microplastic transport will slowly reduce as the model domain is eroded to a flat surface, however we find that as the model resolution increases the flux of microplastic out of the 1D slope converges towards the same trend (Figure 3).**

Table 1:  This Table is confusing. It is not clear that these parameters are varied independently. Initially, I interpreted the Table to indicate that each row indicates a set of linked parameters. But later it became clear that this is not the case. Please make it clearer that the table should be read as three separate tables, not as a series of 3-column rows.

Moreover, it is not clear what the default value is for each parameter when one of the other parameters is varied. Thus, what microplastic grain size and settling velocity is used as the median sediment grain size is varied? Or what are the median sediment size and microplastic grain size as the settling velocity is varied?

**Table 1 has been removed and the description of the models made more precise following the advice below.**

Figures 2, 3: Why are water flux and microplastic flux in area per time, i.e. m2/hr and mm2/hr? Why not volume per time, i.e. m3/hr and mm3/hr? Presumably the 1D slope has unit width, but it would still be more intuitive to interpret the data as volume/time.

**We think it best to keep the units that correspond to a 1D model, m2/hr.**

ln 222-226:  When analysing the impact of the sediment size, which microplastic grain size and fall velocity values did you use for these simulations?

**Details added**

ln 227-232:  When analysing the impact of the microplastic grain size, which median sediment size and which microplastic fall velocity did you use for these simulations?

**Details added**

ln 233-238: When analysing the impact of the microplastic fall velocity, which median sediment size and microplastic grain size did you use for these simulations?

**Details added**

ln 239: Correct typo in "Casear-Lisflood". **Done**

ln 240: Replace "would suggest" with "suggests". **Done**

Figure 4: In caption (c) and (d), replace "microplastic that remains in the active layer" with "microplastic in the active layer". (for simplicity, but also because the active layer in the downstream thalweg may contain microplastics where there were none before – so microplastics were added rather than remaining). **Done**

ln 297: Replace "Institute nationale d'infromation géographique et forestiére", with "Institut national de l'information géographique et forestière". (4 typos in one name, well done) **Thanks!**

ln 301:  Correct typo in "Casear-Lisflood". **Done**

Figure 8: Correct typo in "hgher" in caption. **Done**

ln 320: Correct typo in "bast-fit model". **Done**

ln 324: Replace "however" with "although". **Done**

ln 324: Replace "related for" with "related to" or with "from". **Done**

ln 333: Replace "however" with "but". **Done**

ln 334: Replace "vary" with "varied". **Done**

ln 339: Add reference for "Atmospheric falls could also act as a source of microplastic in soil and along catchment slopes." **Reference added**

ln 349: Replace "Kedzierski et al. (2023)" with "Kedzierski et al., 2023" **Done**

ln 352: Add comma after "polluted soil". **Done**

ln 369: Add "and" after "this model,". **Done**

ln 369: Add comma after "200 m cells". **Done**

Figure 12: Contours are set at uncommon values: 214 m, 414 m, 614 m, … Please drop the 14, and set contours at multiples of 200 m. **Fixed**

Figure 12a: Increase font size of legend label, i.e. "Thickness (m)". **Fixed**

ln 401: Relete "really". **Really? Ok, done.**

ln 420: What is an "addition roughness"? **We have reworded this sentence to explain the additional roughness.**

throughout: Replace "miss-management" with "mismanagement". **Done**

throughout: Check consistency of capitalization of "CAESAR-Lisflood" vs "Caesar-Lisflood". **Done**

Reply to reviewer 2

RC-1. Maybe I missed something, but how do you tell the difference between the sediment and microplastics when estimating the flux or the amount of microplastics that are in the fluvial system, especially if they are the same grain size? Please explain how you separate rock/sediment from the microplastic.

**The microplastics and sediments are tracked as different grains within the model, as defined by their grain size proportion. For each point in the model grid we store the proportion of each grain fraction (MP or other) and then this is updated at each timestep. We propose to improve the methods section to make this accounting clearer.**

**Modified text:**

**(line 123 of track changes file)**

**We will include microplastic as a grain fraction along with a selection of sediment grain fractions, where the input grain size distribution, $F_i$, is distributed over an active layer thickness, $z_a$, to give a thickness of each grain size class, $g_i$ (van de Wiel et al., 2007). Here $i \in \{0, N\}$ and $N$ is the number of grain size classes (including both microplastic and sediment grains). For each grid point in the model, we store the fraction of grains for each class, from microplastic through silts to gravels. The distribution is then updated at each time step as the grains get transported down system by overland flow of water.**

RC-2. When dealing with erosion in the model, how does adding the layer underneath to form the new active layer alter the microplastic distribution? In the model the microplastics go back to zero, but that is not the case in reality.

**We make the simplifying assumption that plastic pollution rests in the top layer of the model and is not buried within the stratigraphy. We can add a constant quantity of MP to the stratal layers, but it would complicate the ability to trace the sources of MP within the river system.**

RC-3. When dealing with deposition in the model, I understand when cutting the active layer, the model transfers the microplastic distribution to the "new" active layer, but how can you justify deleting everything in the "old" layer? The microplastic is still there (in reality) and your measurements would be off, especially if the fluvial environment switches to erosion.

**We did not explain this sufficiently. If the active layer becomes too thick, the lower part becomes frozen in with the grain distribution that it had at the time it became too thick. No MP is lost, simply this new top stratum layer no-longer takes part in the exchange between fluvial erosion and deposition. We can modify the text accordingly.**

**New text:**

**(line 198 of track changes file)**

**Deposition: When the active layer is too thick a new strata layer is created. In this case if the thickness of the active layer exceeds 1.5 times its initial thickness the active layer gets split into a new top strata layer and a new active layer that is 0.5 times the original active layer thickness. The new active layer has the same grain distribution as before and the new top strata layer has the distribution of grains with which it was created frozen in. Upon creation of the new top strata layer, the bottom strata is deleted in the model.**

RC-4. I would suggest making another figure to help explain each model. Like a diagram of the models (e.g. lines190-205).

**We have added a new figure, Figure 2, that is three diagrams of the different model set-ups.**

RC-5. Try to explain how the active layer is defined better. Throughout each model or experiment the stating active layer thickness is changed between 1m, 0.1m, and 0.5m. Is this due to rainfall or slope, and when changing the active layer thickness, how does this change the results, or the comparisons between each model?

**This comment is in line with that raised by Reviewer 1. Active layer thickness was varied to control the limit on the supply of MP for the different model scenarios. This was however never explicitly stated. We will adjust the text to explain this better. We can also run further model scenarios to demonstrate how the active layer thickness influences the model result.**

**We have tried to include the information relating to the active layer thickness in the new Figure 2, and have tried to better explain the variation of this thickness (line 203+ of track changes file)**

RC-6. This may be outside the realm of this research, but did you or have you considered topography, channel geometry, and interactions with channel walls. More or less a 3D model and how would these factors affect the outcomes of the results presented here. How would the values change?

**The model when applied to the Têt has a resolution of 200x200 m, and therefore details such as the channel banks are not captured. At the scale of a river reach these questions could be explored, but at a catchment scale the numerical resolution would be challengingly high if such effects were to be captured. We would therefore suggest that it is indeed outside the realm of this research.**

Line by line comments:

Line 3: change "there is therefore" to "Therefore there is"… **Done**

Line 22: change "of that" to "Of the".. **Done**

Line 23: is there a newer estimate than 2015? **Not really. It is still the reference number**

Line 23: change "and it is estimated" to "and it was estimated" **Done**

Line 44: delete "surprising" **Done**

Line 78: change citation to in sentence format: Wilcock and Crow (2003) **Done**

Line 93: change "rain" to grain  **Done**

Line 174: change "that is there is" to "forming" **Done**

Line 236: delete one of the "the"s, there is two "the" in a row **Done**

Line 242: change "a fork" to something else **"fork" is a technical term for a copy of a github repository.**

Line 270: Change "once" to "one" **Done**

Line 396: delete "between" **Done**

Lines 403-404: how do you know there is this much concentration in the top 50 cm? what if the active layer was errored away. The new active layer is assumed to be at 0? **Changed 50 cm to "active layer (Figure 12a)" where its thickness is plotted**

Line 414: you wrote "density" twice  **Corrected**

Figures 4,6: can you explain what the vertical line in the middle of the model is? these are more visual in Figure 6. In figure 6c it looks like it has been deleted. If this is the active layer, you need to make it more visible. **The vertical line corresponds to the slope break between left- and right-hand side of the 2 slope-model. It is highlighted by the presence or not of microplastic that has either been eroded away or deposited. The colormap is labelled as such. I am unsure how to make this clearer.**

Figure 7. What is the vertical streak of different thicknesses at 5000m? **The thickness of the active layer, as stated in the Figure caption (same comment as just above).**

Line 565: citation not in manuscript **Yes it is (line 46 of track changes file).**

---

## Author Response (AR2)

We would like to thank the editor, Wolfgang Schwanghart, and the associate editor, Daniel Parsons, for helping with our manuscript. We have made the final minor edits required by the reviewer, by adding the reference to line 134 and correcting the final typos.